

# LA Megacity: a High-Resolution Land-Atmosphere Modelling System for Urban CO₂ Emissions

**Sha Feng[1,2*], Thomas Lauvaux[3,2], Sally Newman[4], Preeti Rao[2], Ravan**
**Ahmadov[5,6], Aijun Deng[3], Liza I. Díaz-Isaac[3], Riley M. Duren[2], Marc L.**
**Fischer[7], Christoph Gerbig[8], Kevin R. Gurney[9], Jianhua Huang[9], Seongeun**
**Jeong[7], Zhijin Li[2], Charles E. Miller[2], Darragh O'Keeffe[9], Risa Patarasuk[9],**
**Stanley P. Sander[2], Yang Song[9], Kam W. Wong[4,2], Yuk L. Yung[4]**
[1] JIFRESSE, University of California, Los Angeles, Los Angeles, CA
[2] Jet Propulsion Laboratory, California Institute of Technology, Pasadena, CA
[3] Department of Meteorology, Pennsylvania State University, College State, PA
[4] Division of Geological and Planetary Sciences, California Institute of Technology,
Pasadena, CA
[5] Cooperative Institute for Research in Environmental Sciences, University of Colorado
at Boulder, Boulder, CO
[6] Earth System Research Laboratory, National Oceanic and Atmospheric
Administration, Boulder, CO, USA
[7] Lawrence Berkeley National Laboratory, Berkeley, CA
[8] Max Planck Institute for Biogeochemistry, Hans-Knöll-Str.10, 07745 Jena, Germany
[9] Arizona State University, Tempe, AZ
[*] now at Department of Meteorology, Pennsylvania State University, University Park,
PA 16802, USA
Correspondence to: Sha Feng (sfeng@psu.edu)





## 1  **Abstract**

Megacities are major sources of anthropogenic fossil fuel $CO_2$ emissions. The spatial
extents of these large urban systems cover areas of 10,000 km$^2$ or more with complex
topography and changing landscapes. We present a high-resolution land-atmosphere
modelling system for urban $CO_2$ emissions over the Los Angeles (LA) megacity area.
The Weather Research and Forecasting (WRF)-Chem model was coupled to a very high-
resolution FFCO$_2$ emission product, Hestia-LA, to simulate atmospheric $CO_2$
concentrations across the LA megacity at spatial resolutions as fine as ~1 km. We
evaluated multiple WRF configurations, selecting one that minimized errors in wind
speed, wind direction, and boundary layer height as validated by its performance against
meteorological data collected during the CalNex-LA campaign (May-June 2010). Our
results show no significant difference between moderate- (4-km) and high- (1.3-km)
resolution simulations when evaluated against surface meteorological data, but the high-
resolution configurations better resolved PBL heights and vertical gradients in the
horizontal mean winds. We coupled our WRF configuration with the Vulcan 2.2 (10 km
resolution) and Hestia-LA (1.3-km resolution) fossil fuel $CO_2$ emission products to
evaluate the impact of the spatial resolution of the $CO_2$ emission products and the
meteorological transport model on the representation of spatiotemporal variability in
simulated atmospheric $CO_2$ concentrations. We find that high spatial resolution in the
fossil fuel $CO_2$ emissions is more important than in the atmospheric model to capture $CO_2$
concentration variability across the LA megacity. Finally, we present a novel approach
that employs simultaneous correlations of the simulated atmospheric $CO_2$ fields to
qualitatively evaluate greenhouse gas measurements over the LA megacity. Spatial
correlations in the atmospheric $CO_2$ fields reflect the coverage of individual measurement
sites when a statistically significant number of sites observe emissions from a specific
source or location. We conclude that elevated atmospheric $CO_2$ concentrations over the
LA megacity are composed of multiple fine-scale plumes rather than a single
homogenous urban dome. Furthermore, we conclude that FFCO$_2$ emissions monitoring in
the LA megacity requires FFCO$_2$ emissions modelling with ~1 km resolution since
coarser resolution emissions modelling tends to overestimate the observational
constraints on the emissions estimates.



## 1 Introduction

Carbon dioxide ($CO_2$) is a major anthropogenic contributor to climate change. It has increased from its preindustrial (1750) level of $278 \pm 2$ ppm (Etheridge et al., 1996) to over 400 ppm in recent years, as reported by the National Oceanic and Atmospheric Administration (NOAA) and Scripps Institution of Oceanography [http://co2now.org/]. Clear evidence has shown that the continued increase of the atmospheric $CO_2$ concentration is dominated by global fossil fuel consumption during the same period (IPCC, 2013) and land use change (Houghton, 1999).

Urban areas are significant sources of fossil fuel $CO_2$ ($FFCO_2$), representing more than 50% of the world's population and more than 70% of $FFCO_2$ (UN, 2006). In particular, megacities (cities with urban populations greater than 10 million people) are major sources of anthropogenic emissions, with the world's 35 megacities emitting more than 20% of the global anthropogenic $FFCO_2$, even though they only represent about 3% of the Earth's land surface (IPCC, 2013). The proportion of emissions from megacities increases monotonically with the world population and urbanization (UN, 2006, 2010). Developed and developing megacities around the world are working together to pursue strategies to limit $CO_2$ and other greenhouse gas (GHG) emissions (C40, 2012).

Carbon fluxes can be estimated using "bottom-up" and "top-down" methods. Typically, $FFCO_2$ emissions are determined using "bottom-up" methods, by which fossil fuel usage from each source sector is convolved with the estimated carbon content of each fuel type to obtain $FFCO_2$ emission estimates. Space-time resolved $FFCO_2$ data sets using "bottom-up" methods clearly reveal the fingerprint of human activity with the most intense emissions being clustered around urban centres and associated power plants (e.g., Gurney et al., 2009; Gurney et al., 2012). At the global and annual scale, $FFCO_2$ emission estimates remain uncertain at $\pm 5\%$, varying widely by country and reporting method (Le Quéré et al., 2014). At the urban scale, the uncertainties of $FFCO_2$ emission estimates are often 50-200 % (Turnbull et al., 2011; Asefi-Najafabady et al., 2014). "Top-down" methods could potentially estimate biases in bottom-up emissions, and could also detect trends that cities can use for decision-making, due to changing economic activity or implementation of new emission regulations.





"Top-down" methods involve atmospheric measurements and usually include an
atmospheric inversion of $CO_2$ concentrations, using atmospheric transport models to
estimate carbon fluxes (i.e., posterior fluxes) by adjusting the fluxes (i.e., prior fluxes) to
be consistent with observed $CO_2$ concentrations (e.g., Lauvaux et al., 2012; Lauvaux et
al., 2015; Tarantola, 2005; Enting et al., 1994; Gurney et al., 2002; Baker et al., 2006;
Law et al., 2003). In general, a prior flux is required for estimating the fluxes using an
atmospheric inversion.  The uncertainties in "top-down" methods therefore can be
attributed to errors in the observations (e.g., Tarantola, 2005), emission aggregation
errors from the prior fluxes (e.g., Gurney et al., 2012; Engelen et al., 2002), and physical
representation errors in the atmospheric transport model (e.g., Díaz Isaac et al., 2014;
Gerbig et al., 2008; Kretschmer et al., 2012; Lauvaux et al., 2009; Sarrat et al., 2007).
Previous studies showed that regional high-resolution models can capture the measured
$CO_2$ signal much better than the global models with lower resolution and simulate the
diurnal variability of the atmospheric $CO_2$ field caused by recirculation of nighttime
respired $CO_2$ well (Ahmadov et al., 2009). Pillai et al. (2011 and 2012) and Rödenbeck et
al. (2009) have discussed about the advantages of high resolution CO2 modelling on
different domains and applications. Recent efforts to study $FFCO_2$ emissions on urban
scales have benefited from strategies that apply in-situ observations concentrated within
cities and mesoscale transport models (e.g., Wu et al., 2011; Lauvaux et al., 2015; Strong
et al., 2011; Lac et al., 2013; Bréon et al., 2015).
The Los Angeles (LA) megacity is one of the top three $FFCO_2$ emitters in the U.S. The
atmospheric $CO_2$ concentrations show complex spatial and temporal variability resulting
from a combination of large $FFCO_2$ emissions, complex topography, and challenging
meteorological variability (e.g., Brioude et al., 2013; Wong et al., 2015; Angevine et al.,
2012; Conil and Hall, 2006; Ulrickson and Mass, 1990; Lu and Turco, 1995; Baker et al.,
2013; Chen et al., 2013; Newman et al., 2013). Past studies of exploring $CO_2$
concentrations over the LA megacity used measurement methods ranging from ground-
based to airborne, from in-situ to column. Those studies consistently reported robust
enhancements (e.g., 30-100 ppm in-situ and 2-8 ppm column) and significant variability
of the $CO_2$ concentrations for the LA megacity (Newman et al., 2013; Wunch et al., 2009;
Wong et al., 2015; Kort et al., 2012; Wennberg et al., 2012; Newman et al., 2015). There



have been limited radiocarbon ($^{14}$C) isotopic tracer studies (Newman et al., 2013;
Newman et al., 2008; Djuricin et al., 2010; Riley et al., 2008; Newman et al., 2015).
Newman et al. (2013) showed that FFCO$_2$ constituted 10 - 25 ppm of the CO$_2$ excess
observed in the LA basin by averaging the flask samples at 1400 PST during 15 May –
15 June, 2010. Djuricin et al. (2010) demonstrated that fossil fuel combustion contributed
approximately 50~70 % of CO$_2$ sources in LA. Recently, using CO$_2$ mole fractions and
$\Delta^{14}$C and $\delta^{13}$C values of CO$_2$ in the LA megacity observed in inland Pasadena (2006–
2013) and coastal Palos Verdes peninsula (autumn 2009–2013), Newman et al. (2015)
demonstrated that fossil fuel combustion is the dominant source of CO$_2$ for inland
Pasadena. Airborne campaigns over LA (typically days to weeks in duration) included
ARCTAS-CA (Jacob et al., 2010) and CalNex-LA (Brioude et al., 2013). All of these
earlier studies were limited in their ability to investigate the spatial and temporal
characteristics of LA carbon fluxes given relatively sparse observations. To better
understand and quantify the total emissions, trends, and the detailed spatial, temporal, and
source sector patterns of emissions over the LA megacity requires both a denser
measurement network and a land-atmosphere modelling system appropriate for such a
complex urban environment. In this paper, we couple the Weather Research and
Forecasting (WRF) – Chem model to a high-resolution FFCO$_2$ emission product, Hestia-
LA, to study the spatiotemporal variability of urban CO$_2$ concentrations over the LA
megacity.
The mesoscale circulation over the LA megacity is challenging for atmospheric transport
models due to a variety of phenomena, such as "Catalina" eddies off the coast of southern
California and the coupling between the land-sea breeze and winds induced by the
topography (Angevine et al., 2012; Conil and Hall, 2006; Ulrickson and Mass, 1990;
Kusaka and Kimura, 2004b; Kusaka et al., 2001). In this paper we present a set of
simulations exploring WRF model physics configurations for the LA megacity,
evaluating the model performance against meteorological data from the CalNex-LA
campaign period, 15 May – 15 June 2010. Angevine et al. (2012) also investigated how
WRF model performance varied with spatial resolutions and PBL scheme, etc for the
CalNex-LA campaign period; however, Angevine et al. focused solely on model
meteorological evaluation with spatial resolutions of 12- and 4-km. In the present study



we focus on three critical aspects of the WRF model configuration – the planetary
boundary layer (PBL) scheme, the urban surface scheme, and the model spatial resolution
– as well as the effects of the $FFCO_2$ emissions product spatial resolution. Through these
four aspects, the impacts of physical representation errors and emission aggregation
errors on the modelled $CO_2$ concentrations across the LA megacity are investigated.
Moreover, a novel approach is proposed to evaluate the design of the greenhouse gas
(GHG) measurement network for the LA megacity. The LA measurement network
consists of 15 observation sites designed to provide continuous atmospheric $CO_2$
concentrations to assess the anthropogenic carbon emissions distribution and trends. The
goal of the network design exploration is to optimize the atmospheric observational
constraints on the surface fluxes. Kort et al. (2013) found that a minimum of eight
optimally located in-city surface $CO_2$ observation sites were required for accurate
assessment of $CO_2$ emissions in LA using the "footprint" method (backward mode) and
based on a national $FFCO_2$ emission product Vulcan. Here we assess the influence of
each observation site using spatial correlations in terms of the simulated $CO_2$ (forward
mode) at high-resolution.
The remainder of the paper is organized as follows. Section 2 describes the modelling
framework, including initial conditions and boundary conditions for WRF-Chem.  In
section 3, we assess the quality of the model results, focusing on accurate representation
of the PBL height, wind speed and wind direction.  Section 4 presents the spatial and
temporal patterns of simulated $CO_2$ concentration fields over the LA megacity using
various $FFCO_2$ emissions products.  Section 5 describes the forward mode approach for
evaluating the spatial sensitivity of the 2015-era surface GHG measurement sites within
the LA megacity. Discussion of model errors, model sampling strategy, and the density of
the LA GHG measurement network from the forward model perspective is given in
section 6. A summary is given in section 7. Section 8 lists the author contributions.
**2   Modelling Framework**
Sensitivity experiments were conducted using WRF-Chem version 3.6.1 with various
PBL schemes, urban surface schemes, and model resolutions to define an optimized





configuration for simulating atmospheric $CO_2$ concentration fields over the LA megacity.
The impact of the resolution of $FFCO_2$ emission products is investigated as well.

## 2.1  WRF model setup

All of the model runs used one-way triple-nested domains with resolutions of 12-, 4-, and
1.3-km. The coarse domain (d01) covers most of the western US; the intermediate
domain (d02) covers California and part of Mexico (Figure 1a); the innermost domain
(d03) covers the majority of the South Coast Air Basin, a portion of the southern San
Joaquin Valley and extends into the Pacific Ocean to include Santa Catalina and San
Clemente Islands (Figure 1b). The Los Angeles basin is surrounded to the north and east
by mountain ranges with summits of 2-3 km, with the ocean to the west and the desert to
the north. The basin consists of the West Coast Basin, Central Basin, and Orange County
Coastal Plain. The boundaries of these three regions are Newport Inglewood Fault and
the boundary between Los Angeles County and Orange County. In this study, our
analysis is limited to the innermost domain (d03), referred to hereafter as the LA
megacity. All three of the model domains use 51 terrain following vertical levels from
surface to 100 hPa, of which 29 layers are below 2 km above ground level (AGL).
The meteorological fields and surface parameters, such as soil moisture, were initialized
by the three-hourly North American Regional Reanalysis (NARR) data set with a
horizontal resolution of 32 km (Mesinger et al., 2006) and the six-hourly NCEP sea
surface temperature data set with a horizontal resolution of 12 km
(ftp://polar.ncep.noaa.gov/pub/history/sst/ophi). A summary of WRF configurations
common to all sensitivity runs is shown in Table 1. The impact of varying the PBL
parameterization, urban surface, and model resolution was investigated by conducting
sensitivity runs summarized in Table 2.
PBL schemes are used to parameterize the unresolved turbulent vertical fluxes of heat,
momentum, and constituents within the planetary boundary layer. There are tens of
mesoscale PBL schemes available in the WRF package. We selected the three most
commonly used turbulent kinetic energy (TKE)-driven PBL schemes for the sensitivity
runs: the Mellor-Yamada-Janjie technique (MYJ, Janjić, 1994), Mellor-Yamada




Nakanishi and Niino Level 2.5 (MYNN, Nakanishi and Niino, 2006), and Bougeault-
Lacarrère (BouLac, Bougeault and Lacarrere, 1989). The TKE-driven PBL schemes
explicitly estimate the turbulent fluxes from mean atmospheric states and/or their
gradients and can be used to drive a Lagrangian particle dispersion models in subsequent
atmospheric inversions (e.g., Lauvaux et al., 2008).
For an accurate representation of the LA $CO_2$ simulation, the necessity of incorporating
the urban surface scheme was tested by alternatively including an urban canopy model
(UCM, Kusaka and Kimura, 2004a), a building environment parameterization (BEP,
Martilli et al., 2009), and no urban surface scheme.
We chose to test and evaluate our WRF-Chem configuration during the May-June 2010
time period of the CalNex-LA campaign (Ryerson et al., 2013) to take advantage of the
extra meteorological measurements recorded during the campaign. Hourly simulations
were conducted for 36-h periods starting with a 12-h meteorological spin-up at 12:00
UTC of the previous day. Hence, when concatenating the model output, each new run is
introduced at 0000 UTC. All of the analyses in the following sections are limited to the
region of the LA megacity.

## 2.2    Configuration for the $CO_2$ simulation

WRF-Chem version 3.6.1 was modified to allow for online $CO_2$ tracer transport coupled
with the Vegetation Photosynthesis and Respiration Model (VPRM) (Ahmadov et al.,
2007; Xiao et al., 2004). VPRM calculates hourly net ecosystem exchange based on
MOIDS satellite estimates of the land surface water index and enhance vegetation index,
short wave radiance and surface temperature. A detailed description of VPRM can be
found in Mahadevan et al. (2008).
Anthropogenic $FFCO_2$ fluxes were alternatively prescribed from the Vulcan 2.2 and
Hestia-LA 1.0 $FFCO_2$ emission products developed at Arizona State University (Gurney
et al., 2009; Gurney et al., 2012; Gurney et al., 2015; Rao et al., 2015). Both emission
products were developed using "bottom-up" methods. Vulcan quantifies $FFCO_2$
emissions for the entire contiguous United States (CONUS) hourly at approximately 10
km spatial resolution for the year of 2002, combining data sources such as local pollution



reporting, traffic data, and point source monitoring (Gurney et al., 2009). Hestia-LA, by
contrast, is a fossil fuel $CO_2$ emissions data product specific in space and time to the
individual building, road segments, and point sources of the Los Angeles megacity (Rao
et al., 2015; Gurney et al., 2015; Gurney et al., 2012; Zhou and Gurney, 2010).
Leveraging from the Vulcan constraint at the county level, Hestia-LA quantifies $FFCO_2$
emissions for Los Angeles County, Orange County, San Bernardino County, Ventura
County, and Riverside County, at approximately 1.3 km x 1.3 km every hour of the years
of 2011 and 2012.  More details about Hestia-LA see Rao et al. (2015).
Atmospheric $CO_2$ concentrations in WRF-Chem were alternatively driven by the Vulcan
and Hestia-LA emissions at the resolutions of 4 km and 1.3 km. Hence, four different
emission datasets were generated – Vulcan 10 km emissions transported at 4-km or 1.3-
km resolution, and Hestia-LA 1.3 km emissions transported at 4-km or 1.3-km resolution.
The Hestia-LA emissions were aggregated from the native building-level resolution to
the 1.3 and 4 km resolutions via direct summation in the specified model grids. Hestia-
LA 2011 is temporally shifted for creating the weekday-weekend cycle for the year of
2010. The Vulcan $FFCO_2$ emissions were interpolated by using a bilinear operator and by
preserving the value of the integral of data between the source (10-km) and destination
(4- and 1.3-km) grid. Also, the ratio of the total carbon emissions over the state between
the years of 2002 and 2015 from California Air Resource Board (http://www.arb.ca.gov/)
was uniformly applied to the Vulcan emissions to temporally scale Vulcan from the 2002
base year to 2010. At regional scales, anthropogenic and biogenic fluxes are much larger
than ocean fluxes. Hence, no $CO_2$ ocean fluxes were prescribed. This paper analyses the
impact of both physical representation errors and emission aggregation errors on the
modelled $CO_2$ concentrations across the LA megacity.
Lateral boundary conditions and initial conditions for $CO_2$ concentration fields were
taken from the three-dimensional $CO_2$ background (often called "NOAA curtain" for
background) estimated from measurements in the Pacific (Jeong et al., 2013).



## 3    Model – data comparison
Meteorological    observations    obtained    during    the    CalNex-LA    campaign
(http://www.esrl.noaa.gov/csd/projects/calnex/) include PBL height sampled by NOAA
P-3 flights and aerosol backscatter ceilometer (Haman et al., 2012; Scarino et al., 2013), a
radar wind profiler operated by the South Coast Air Quality Management District near
Los Angeles International Airport (LAX), and $CO_2$ in situ measurements (Newman et al.,
2013). Additionally, the NWS (National Weather Service, www.weather.gov) surface
observations are used.
### 3.1    Comparison to aircraft PBL height
During CalNex-LA, 17 P-3 research flights sampled the daytime and nighttime PBL,
marine surface layer, and the overlying free troposphere throughout California (Ryerson
et al., 2013).   We imposed five criteria for selecting aircraft profiles of potential
temperature for PBL height comparisons:
1)  Aircraft profiles sample within the innermost model domain (d03, Figure 1b);
2)  Profiles sample during daytime (1100 PST – 1700 PST) when the $CO_2$ concentrations

16        in PBL is well mixed;

3)  Profiles acquired within ±30 min of the model output;
4)  Profiles with valid sampling below and above 1 km AGL to assure the chance to

19        determine the PBL height from the potential temperature vertical gradient;

5)  Ability to determine the PBL height from the vertical gradient of potential

21        temperature.

Based on these five criteria, we selected seven aircraft profiles collected between 16 May
and 19 May 2010. Figure 2 shows a profile acquired on 19 May 2010 when the aircraft
was sampling over Pasadena.
The model diagnostic PBL height calculated by each PBL scheme can differ from the
others due to the Richardson bulk number ($R_i$) used (e.g., Kretschmer et al., 2014; Hong
et al., 2006; Yver et al., 2013). To avoid this difference, we determined modelled PBL
height based on the vertical virtual potential temperature gradient. For the case (Figure
2), the modelled PBL height agrees within 50 meters of the aircraft-determined and
ceilometer-measured PBL height
Figure 3 shows the absolute difference between the modelled and aircraft-determined
PBL height for each selected aircraft profile. The differences between the modelled and
aircraft-determined PBL height differ case by case. None of the model physics is
systematically better than others. However, BouLac_BEP and MYNN have larger biases
than others. The averaged bias of BouLac_BEP is 289 m for d02, 295 m for d03; MYNN
is 179 m for d02 and 216 m for d03. For other configurations, the averaged biases are
smaller than 160 m. The modelled PBL bias appears somewhat smaller in the 4-km runs
than the 1.3-km runs. This, however, is obtained based on seven selected aircraft profiles
only. To further define the optimal physics for the PBL height simulation, we will present
the all-hours statistics with the ceilometer data in section 3.2.

### 3.2   Comparison to ceilometer PBL height

Accurate simulation of the time evolution of the PBL height is crucial to properly
simulate the vertical mixing and ventilation of $CO_2$ emitted at the surface. The ceilometer
measurements during CalNex-LA (Haman et al., 2012) allow us to evaluate the time
evolution of the modelled PBL height. Compared with the ceilometer-measured PBL
height, the maximum discrepancies between model and observations occur from around
1100 PST – 1200 PST when the nocturnal PBL is fully collapsed and 1700 PST when it
starts to form again (Figure 4). Among all of the model physics, MYNN_UCM shows the
best agreement with the observations, while BouLac_BEP differs from ceilometer the
most.  The absolute bias of the MYNN_UCM modelled PBL height ranges from 5 to 198
m and 0 to 184 m with mean bias of -15.3 ± 66.1 m and -6.9 ± 82.7 m for d02 and d03,
respectively, suggesting the 1.3-km model resolution statistically improves the model
performance in the PBL simulation as compared with the ceilometer. The improvement
in the high-resolution model runs can be seen in other configurations as well. However,
the ceilometer measurements were all at Caltech and thus reflect interior conditions.
These are expected to be very different from coastal conditions in terms of the temporal
evolution and eventual height of the mid-day PBL as well as the timing of the nocturnal



PBL collapse, etc. The domain is much larger and more varied than captured by a single
location.
We also notice that using UCM-coupled simulations agree with the ceilometer better than
other combinations (MYNN_UCM vs. MYNN, MYJ_UCM vs. MYJ, BouLac_UCM vs.
BouLac_BEP). Using UCM can largely reduce the difference across the model runs and
discrepancy from the observations.
**3.3   Comparison to radar wind profiler**
Atmospheric dynamics has a direct influence on the $CO_2$ transport. Realistically
reproducing the vertical gradient of wind fields is crucial. In Figure 5, we show the
average difference in the wind profiles between the models and the radar wind profiler at
LAX (Angevine et al., 2012). Most of the simulations show relatively larger wind speed
bias near the surface: BouLac_BEP, MYJ, and MYNN with bias of 2.4 ± 2.2 m/s,
BouLac_UCM and MYJ_UCM with bias of 2.0 ± 2.3 m/s. In contrast, it is encouraging
to see that MYNN_UCM agrees with the radar measurement best with mean bias of 1.4 ±
2.0 m/s, a lower mean bias than for the other configurations. Additionally, UCM-coupled
simulations tend to reduce the wind speed bias at this location.
For wind direction, likewise, MYNN_UCM agrees with the observations slightly better
below 800 m (About 1.1 m/s for the averaged error), although the model bias is much less
pronounced across the configurations. However, we notice that MYNN_UCM shows
larger wind direction bias between 800 – 1400 m than others due to relatively lower PBL
height simulated (not shown).
Improvement provided by the 1.3-km model resolution is visible near the PBL height
(800 – 1400 m). A finer model resolution tends to resolve the vertical gradients of the
atmospheric states better. This also can be demonstrated by the PBL comparisons with
ceilometer (Figure 4).
Angevine et al. (2012) evaluated a set of model configurations with the highest model
resolution at 4 km for CalNex-LA using the same radar wind profiler data. The optimal
configuration (the total energy–mass flux boundary layer scheme and ECMWF
reanalysis) they found showed 1.1 ± 2.7 m/s bias in wind speed and -2.6 ± 67° in wind





direction near the surface. Here MYNN_UCM displays similar performance to the
optimal configuration they concluded.   At the 4-km model resolution, the biases of
MYNN_UCM are 1.4 ± 2.0 m/s in wind speed and -1.3 ± 20.0° in wind direction. In
section 3.4, we will examine the performance of MYNN_UCM across the LA megacity.
**3.4   Comparison to NWS surface stations**
Due to the limited number of observation sites available at this time in this region, the
analysis above can only be done at specific locations. We therefore introduce the NWS
surface network to demonstrate the model performance across the LA megacity. The
objective analysis program OBSGRID is used to remove erroneous data and observations
that are not useful (Deng et al., 2009; Rogers et al., 2013).
Figure 6 shows the model bias compared to the NWS surface data across the LA
megacity. The locations of the GHG measurement sites are marked (see details in Table 3
and Figure S1). Overall, there is little difference in the simulated surface atmospheric
state variables between the 4-km and 1.3-km runs; i.e., the 1.3-km run does not show any
significant improvement compared to the 4-km run at the surface (even though it resolves
the vertical gradient of atmospheric states and PBL better, Figure 4 and 5).
For temperature (Figure 6a1 and 6b1), the model is colder than the observations by 0.5 -
1.0 K.  Larger temperature biases occur in the desert. For relative humidity (Figure 6a2
and 6b2), the model is dry compared to the observations, which is consistent with the
findings of Nehrkorn et al. (2012). The model is 5% dryer over the basin with a
somewhat larger bias of 5% - 10% near Granada Hills and Ontario that have the highest
temperature in the summer – typically 20 °F or more warmer than downtown LA in May-
June. The dryness in the model tends to cause lower PBL heights, which can be seen in
the comparison to the ceilometer-determined PBL height at Caltech in Pasadena,
California (Figure 4): MYNN_UCM shows a shallower PBL in comparison to the
ceilometer during the 1400 PST – 1800 PST time period.
The model overestimates wind speed by ~1.0 m/s (Figure 6a3 and 6b3). The tendency of
the model to overestimate wind speed is fully documented in previous studies (e.g.,
Angevine et al., 2012; Brioude et al., 2013; Nehrkorn et al., 2012; Yver et al., 2013). For





surface wind direction, model bias is within ±10° for most of the LA megacity. The
larger biases appear near the foothills of Santa Monica Mountains, San Gabriel
Mountains, and University of Southern California (USC) due to the challenging land
surface and terrain.
Compared with other model physics (not shown), we notice that USC located in the
downtown LA is a challenging location for mesoscale modelling, in particular for wind
simulations. All of the model physics consistently show a relatively large wind bias at
USC except BouLac_BEP that fails in the remainder of the domain. We also noticed that
adding UCM to MYNN decreases the modelled temperature, while all of other models'
physics have a warm bias compared to observations.
All of the analyses above focused on the meteorology over the LA megacity. The results
indicate little difference horizontally between 4- and 1.3-km runs across the basin, which
is consistent with the Angevine et al. (2012) assumption that a finer grid may not give
better results. However, the 1.3-km run tends to resolve the vertical gradients of
atmospheric state variables and PBL better, which can improve the vertical mixing and
ventilation of modelled atmospheric $CO_2$ concentrations.
Overall, the MYNN_UCM configuration showed the best agreement with meteorological
observations of all the configurations we evaluated. Therefore, we will use the MYNN-
UCM configuration in our simulations of atmospheric $CO_2$ concentration fields over the
LA megacity.
**3.5   Comparisons to in-situ $CO_2$**
We coupled Hestia and Vulcan $FFCO_2$ emission products individually with the
MYNN_UCM WRF  configuration to generate four sets of simulated $CO_2$ concentrations:
WRF-Hestia 1.3-km, WRF-Hestia 4-km, WRF-Vulcan 1.3-km, and WRF-Vulcan 4-km.
The runs with the same model resolution have the same meteorology but differ in
emissions, and vice versa.
During CalNex-LA, in-situ observation sites at Pasadena and Palos Verdes continuously
measured surface $CO_2$ concentrations. Measurements were recorded using a Picarro
(Santa Clara, CA) Isotopic $CO_2$ Analyser (cavity ring-down spectrometer), model G1101-



i, for Pasadena and an infrared gas analyser from PP Systems (Haverford, MA), model
CIRAS-SC for Palos Verdes. In addition, periodic flask samples were collected for
analysis of $^{14}CO_2$ for extracting fossil fuel and biogenic signals. See Newman et al.
(2015) for details about the sites and sampling information. Figure 7 shows the
comparison of the time series of hourly (Figure 7a,b) and daily afternoon (Figure 7c,d)
averaged $CO_2$ concentrations (1300 PST – 1700 PST) between model and observation.
Overall, the model captures the temporal variability of $CO_2$ but overestimates $CO_2$ during
nighttime. During afternoons, the model agrees with the observations fairly well (Figure
7c and 7d) except for a few events: all simulations underestimate $CO_2$ concentrations by
about 10 ppm around May 28 and June 4-6 for Pasadena and May 21 for Palos Verdes.
These events lasting two – three days are likely related to synoptic scale processes. Using
the averaged Pacific Ocean $CO_2$ signal as background may explain the failure to capture
these events. Further investigation of the background air would provide insights related to
synoptic variability but is beyond the scope of this work. We focus here on the diurnal
variability.
Clear diurnal variations of the surface $CO_2$ concentrations were observed for both sites
(Figure 8). The observed $CO_2$ concentrations increase at night and remains high until
sunrise, and quickly drop as the boundary layer grows after sunrise (Figure 8a and 8b).
For the Pasadena site, during nighttime, when the PBL is shallow, $CO_2$ is trapped locally:
the more fossil fuel is emitted, the higher $CO_2$ concentration is simulated. Consequently,
the WRF-Vulcan runs show considerably lower $CO_2$ concentration than the WRF-Hestia
runs due to the lower emissions in Vulcan at the Pasadena site (Figure 8c). However,
during daytime, with well-mixed conditions, the discrepancy between the WRF-Hestia
and WRF-Vulcan runs becomes smaller. Among these runs, the 1.3-km WRF-Hestia run
successfully captures the diurnal variation of the surface $CO_2$ concentration, although a
peak is not present in the observation around noon. By contrast, the 4-km WRF-Hestia
run underestimates the $CO_2$ concentration during 0200 – 0700 PST even though
emissions were comparable between Hestia 4-km and Hestia 1.3-km (Figure 8c). The
underestimation of the simulated $CO_2$ concentration must mainly result from the
representation errors in the atmospheric transport due to the coarser model resolution.





For Palos Verdes, however, none of the model results match the observations. All of the runs show a peak in the simulated $CO_2$ concentration around 0800 PST, which very likely corresponds to the eastward marine flow as a part of the Catalina eddy (e.g., Bosart, 1983; Davis et al., 2000). This $CO_2$ concentration peak is incorrectly reproduced by the model advecting the $FFCO_2$ emitted from the strong point sources in Long Beach, California (Figure 1d) and in turn contaminating the air of Palos Verdes.

## 4   Spatial pattern of the surface $CO_2$

The spatial pattern of surface $CO_2$ concentration exhibits diurnal variability over the LA megacity due to the complexity of the topography and the variability of circulation patterns, PBL heights, and $FFCO_2$ emissions. Each plays an important role in sequence or at the same time. Here, we only focus on the pattern at 1400 PST when the atmospheric $CO_2$ concentration is well mixed in the PBL. At 1400 PST, there is a close relationship between $CO_2$ concentration and atmospheric transport; the error due to the PBL height determination is at a minimum. For the same reason, we show that $FFCO_2$ emissions do not play a dominant role around 1400 PST unless there are strong local signals from point sources, such as power plants, refineries, airports etc.

In this section, we define the 1.3-km WRF-Hestia run as the reference simulation. For simplicity, all of the relevant $CO_2$ spatial patterns we present are selected from the second model layer (about 24 m AGL). Figure 9a and 9b display the topography and the average $CO_2$ concentration at 1400 PST overlaid with the first empirical orthogonal function (EOF1) of the surface wind pattern, respectively. The locations of the 13 GHG measurement sites in the LA megacity domain are marked in the figures (see Table 3 and Figure S1 for details about the observation sites). Note that The 2015-era surface GHG measurement network includes 14 sites in total, while 13 sites are included in the innermost model domain. According to the geography mentioned in section 2.1, the Granada Hills (GH), Compton, USC, and sites are located in the West Coast Basin, the Pasadena and Mt. Wilson (MWO) sites are in the Central Basin, and California State University Fullerton (CSUF), Ontario, and San Bernardino (SB) sites are in the Orange County Coastal Plan. Additionally, the Dryden and Victorville (VV) sites are located in





deserts; the Palos Verdes (PV), University of California Irvine (UCI), and San Clemente
Island (SCI) are on the coast. Although the Dryden site is actually a TCCON site, in the
analysis, we assume it is a near-surface point measurement like other sites for simplicity.
Blocked by the mountains, the emitted $CO_2$ is trapped in the basin; the desert is as clean
as the upwind ocean. Specifically, Dryden (not shown on the figure), VV, SCI (not
shown on the figure), Palos Verdes and UCI are much cleaner than other sites (Figure
9b). At 1400 PST, sea breeze prevails over the LA megacity. Affected by the geometry of
Palos Verdes Peninsula, the sea breeze is divided into west and southwest onshore flows
and then converge in the Central Basin. Strong $CO_2$ signals emitted from electricity
production and industry (with annual emission of 86.9 million kgC, Figure 1d) are
trapped in a limited area. We notice that the south-western flow, which appears stronger
than the western flow, prevents the high $CO_2$ concentration in the West Coast Basin from
propagating further east and dilutes into the Central Basin. Controlled by the orography,
strong southerly flows occur between the Santa Monica and San Gabriel Mountains,
keeping the contaminated air from propagating to the west. Driven by the same
meteorology, the 1.3-km WRF-Vulcan run shows a more smeared out $CO_2$ concentration
over the LA megacity (Figure 9c) due to the coarser resolution of the original Vulcan
emissions. High $CO_2$ plumes seen in the 1.3-km WRF-Hestia run from point sources are
replaced by wide area of the elevated $CO_2$ concentration in the 1.3-km WRF-Vulcan. The
large differences in the simulated surface $CO_2$ fields between the 1.3-km WRF-Hestia and
WRF-Vulcan runs are around LAX and north of the Palos Verdes Peninsula where strong
point sources are located (dipole-like pattern in Figure 9d).

## 24   5   Sampling density of the 2015-era GHG measurement network

In this section, we present a forward network design framework, using the modelled $CO_2$
concentrations and their relationship with neighbouring grid cells. Compared to previous
studies using tower footprints (i.e. linearized adjoint models) as Kort et al. (2013), we
propose here a forward model assessment of the network using our high-resolution WRF
results. We assume that each observation site can be associated with a specific $CO_2$ air
mass at any given time. To define this $CO_2$ air mass, we estimate the spatial coherence in





the modelled $CO_2$ concentration fields. We constrain the coverage of each LA GHG
measurement site by calculating the simultaneous correlation of the site to the rest of the
domain using the simulated $CO_2$ concentration time series. Figure 10 shows the
correlation map (R) of each site for the 1.3-km WRF-Hestia run. Only areas meeting a
significance level of 0.01 in the t-test ($|R| \geq 0.46$) are coloured. Based on the spatial
patterns of the correlation maps, all of the observation sites can be grouped into (i)
coastal/island sites, i.e., UCI, SCI, and Palos Verdes (right three panels in bottom row of
Figure 10), (ii) western basin sites, i.e., GH, Pasadena, MWO, USC, and Compton (top
row in Figure 10), (iii) eastern basin sites, (i.e., CSUF, Ontario, SB; middle row in Figure
10), and (iv) desert sites, i.e., Dryden and VV (left two panels in bottom row of Figure

11    10).

Not surprisingly, the coastal/island sites are mainly correlated with $CO_2$ concentration in
upwind areas offshore where there is limited $FFCO_2$ contamination. The white channel
from Catalina Island to the Huntington Beach area demonstrates the influence of terrain-
induced flows and mountain blocking. The western basin sites are mainly correlated with
$CO_2$ concentration throughout the western portion of the basin, and the eastern basin sites
are mainly correlated with $CO_2$ concentration throughout the eastern portion of the basin.
The desert sites are anti-correlated with the basin. CSUF also shows anti-correlation with
the desert. Two reasons can explain this anti-correlation. Firstly, $CO_2$ is trapped and
accumulates in the basin due to the mountain barrier. Secondly, after $CO_2$ accumulates in
the basin over a certain amount of time, episodic strong sea breezes may push this basin
$CO_2$ over the mountains to the desert. As a result, the basin will be relatively clean while
the desert is contaminated.
Based on the correlation maps, we can also see how the coverage of each site varies with
the $FFCO_2$ emissions data products and with the WRF model resolutions. Figure 11
shows the correlation maps across the runs for the Compton, Palos Verdes, and CSUF
stations. All runs use the optimal physics we determined for the LA megacity, i.e.,
MYNN_UCM. The correlation maps for each site differ with the $FFCO_2$ emissions data
product used, model resolution, or their combination (Figure 11). Given that the 1.3-km
WRF-Hestia is the reference run, the difference of this to the 1.3-km WRF-Vulcan run
reflects the errors induced by emissions resolution. The difference between the 4-km




WRF-Hestia run and the 1.3-km WRF-Hestia run reflects by the model representation
errors. The 4-km WRF-Vulcan run is subject to model representation errors and emission
aggregation errors at the same time. For simplicity, we will not emphasize but show the
comparison of the 4-km WRF-Vulcan to the others.
Compton is isolated from the rest of the basin in the 1.3-km WRF-Hestia run but
correlated with most of the basin in the 1.3-km WRF-Vulcan run. A similar discrepancy
is seen for Palos Verdes.  Additionally, Palos Verdes appears to be a clean site in the 1.3-
km WRF-Hestia run but dramatically contaminated in the 1.3-km WRF-Vulcan run (even
correlated with the LA downtown area). For CSUF, the anti-correlation between basin
and desert noted above is not visible in the 1.3-km WRF-Vulcan run. Compared to the
1.3-km WRF-Hestia run, the 4-km WRF-Hestia run overall shows a somewhat larger
region with significant correlation for each site.
To highlight the discrepancy of the spatial pattern caused by the model representation
errors and emission aggregation errors in the view of the existing GHG measurement
network, a composite map for each run is shown in Figure 12. These maps are
constructed by determining the number of sites for which the absolute value of R is
greater than 0.46 for each grid cell (i.e., colour-filled area in Figure 10 and 11). R=0.46 is
the critical value for the *t*-test at the significance level of 0.01. In the 1.3-km WRF-Hestia
run (reference), the West Coastal Basin and Orange County Coastal Plain are correlated
with up to 6 measurement sites. A gap appears over the Central Basin correlated with up
to 3 sites due to the wind pattern (Figure 9a and 9b). The San Gabriel Mountains and
Peninsular Ranges are rarely correlated to any of the sites due to the elevated terrain. The
4-km WRF-Hestia run shows a similar pattern but with more sites covered over the
Peninsular Ranges and the coast because of the failure to resolve topography by the 4-km
model resolution.
In the 1.3-km WRF-Vulcan run, by contrast, a large area of the basin is correlated with
most of the sites (9 sites out of 13). The Compton area is even correlated with 11 sites,
which is only correlated with about two sites in the 1.3-km WRF-Hestia run. A similar
contrast can be seen for the GH, USC, and Palos Verdes areas where the multiple strong
point sources nearby in Hestia-LA have been aggregated into one 10 km by 10 km grid



cell in Vulcan (Figure 1d vs.1c). Relatively coarser FFCO$_2$ emissions artificially increase
the coverage of each site, which highlights the importance of using Hestia for the CO$_2$
simulation for urban environment to represent the spatial variability in CO$_2$ and design
the optimal network of surface GHG measurement.
**6   Discussion**
Isotopic tracer radiocarbon ($^{14}$C) can be used for distinguishing between fossil fuel and
biogenic sources of CO$_2$ (Djuricin et al., 2010; Newman et al., 2013; Newman et al.,
2008; Pataki et al., 2006; Pataki et al., 2007; Levin et al., 2003; Miller et al., 2012;
Turnbull et al., 2006; Turnbull et al., 2009). During CalNex-LA, two-weeks' flask
samples were combined to produce two CO$_2$ samples for extracting anthropogenic and
biogenic signals from the total CO$_2$ concentration. Note that the two samples for Palos
Verdes were sampled from 1 May to 31 May and from 1 June to 30 June, not exactly
overlapping the CalNex-LA period; the two for Pasadena were sampled from 15 May to
31 May and from 1 June to 15 June, overlapping the CalNex-LA period. See Newman et
al. (2015) for details about the sites and sampling information. Figure 13 presents the
comparisons of the modelled and flask-sampled anthropogenic fossil fuel and biogenic
CO$_2$. From both the flask samples and model simulations, the CO$_2$ signal from the
biosphere is much weaker than FFCO$_2$ in the LA megacity. The two-week flask sampled
biogenic CO$_2$ is about 2 ppm on average. We notice that the 1.3-km WRF-Vulcan
overestimates the FFCO$_2$ concentrations about 20 ppm over the second half of the month
(Figure 13d), implying that low-resolution CO$_2$ emissions can be very critical for a coast
site (complex terrain) with strong point source nearby.
Strong temporal variability of the simulated biogenic and FFCO$_2$ can be seen for both
sites (Figure 13a,13c,13e,13g). For the Pasadena site, the 1.3-km run shows nearly flat
biogenic CO$_2$ concentrations during 15 May to 30 May when the 4-km run has more
variability (Figure 13e). We notice that a large botanical garden covering 207 acres (i.e.
The Huntington Library) is about 1.6 km away from the Pasadena site, which may
suggest that higher model resolution (1.3 km vs. 4 km) could be impacted by a change in
land cover. However, there is still up to about 3-ppm discrepancy in the modelled



biogenic $CO_2$ from the flask samples (Figure 13f). Similar discrepancy can be seen for
Palos Verdes as well (Figure 13h). Reasonably determining $CO_2$ from biogenic sources
remains challenging. Additional measurements are needed to constrain biogenic fluxes.
Here, we focus on $FFCO_2$ emissions that dominate local $CO_2$ signals across the basin.
The results presented in this paper have shown that the choice of model resolution and
emission products can strongly influence the interpretation of atmospheric $CO_2$ signals.
Hestia quantifies $FFCO_2$ emissions down to individual buildings and roadways, in which
strong point sources create large plumes that are extremely sensitive to atmospheric
transport. Reproducing dynamics realistically by the atmospheric transport model is
crucial around strong point sources, such as power plants, refineries, airports, etc. For
instance, a considerable number of point sources are located in Long Beach  (harbours,
Figure 1d), about 7 km away from Palos Verdes. In late spring and summer, Palos Verdes
is a clean site, with little evidence of $FFCO_2$ emissions from the LA megacity most of the
time. However, we can clearly see oftentimes Palos Verdes is simulated to be
contaminated by $FFCO_2$ in all of the runs, especially during early morning (Figure 8b)
due to incorrectly simulated east marine flows advecting the strong $FFCO_2$ emissions,
which cannot be seen in the observations. Bias in wind speed and direction becomes
critical for such a location. Palos Verdes may be challenging for the atmospheric model if
used as a background site.
For a location like Compton with strong point sources nearby emitting $CO_2$  at 86.9
million kgC per year (recorded in Hestia-LA version 1.0), a fine resolution emission
product becomes very important due to the strong $FFCO_2$ gradient. A relatively coarse
emission product likely produces a spurious signal due to aggregating a strong point
source into a large grid cell (Figure 9b and 9c). For instance, dipole-like $CO_2$ gradients
were created in the difference between the 1.3-km WRF-Vulcan and WRF-Hestia runs
(Figure 9d).
In this paper, we focus on the spatial distribution of the $CO_2$ concentration over the LA
megacity. The choice of model resolution also significantly impacts the vertical gradients
of the $CO_2$ concentration as a result of the terrain resolved.  The 1.3-km model runs
approximates the elevation of MWO as 1129 m, while the 4 km runs is 753 m; the actual





elevation is 1600 m. The representation errors in the 4-km model resolution are relatively
large. When there is better topographic resolution, more $CO_2$ is accumulated in the basin
due to blocking by the mountains.    Around noon, the model results show $CO_2$
enhancement of 10 ppm over MWO in both of the 1.3-km WRF-Vulcan and WRF-Hestia
runs but only up to 3 ppm in the 4-km model runs. Additionally, because of the reasons
above, reasonable sampling strategy is worth investigating for the mountain sites like
MWO (e.g., Law et al., 2008). Similar problems exist for a site like Palos Verdes, since
the coastline resolved varies with the model resolutions, as does the topography. Model
sampling strategy is therefore recommended even at 1.3-km resolution, as no clear
improvement in the meteorological evaluation was observed in horizontal.
Figure 10 presents the simultaneous correlation maps for each site in terms of the
simulated $CO_2$ concentration time series. The coverage of the correlation maps is
determined by two factors at the same time: atmospheric transport and surface fluxes.
This method differs from the footprint method (Kort et al., 2013). The footprint method
indicates the influence of the atmospheric transport to the location of the observation
only; no emission pattern was considered. Here both transport and emissions play a role
in the area covered by the observation site. Therefore, the correlation maps are subject to
overestimation of the influence area versus the footprint method, due to the complicated
nature of the atmospheric integrator. As an example, in Figure 10, the coloured grids of
the correlation map are not necessarily *physically* related to the observation site.  Those
far from the site may lose the track of the initial sources. Conversely, there is definitely
no *physical* influence from the uncorrelated areas to the observation site. Figure 14 shows
the fraction of the total $FFCO_2$ emissions over the LA megacity as function of the number
of the observation sites for all of the runs.  Because of the reason above, we focus on the
uncorrelated areas only. Assuming that the coverage of the GHG measurement network is
not sufficient if an area is correlated to less than or equal to two sites, then ~28.9 % of
$FFCO_2$ is potentially under-constrained by the current GHG measurement sites (Figure
14a: WRF-Hestia 1.3-km). These areas include most of the mountains, Santa Monica Bay
and the upwind coast, and the south part of the Central Basin (Figure 11), about 21.1 %
of total area. However, this analysis is a qualitative assessment of the observational



constraint. Consideration of errors in the $CO_2$ emissions needs to be taken into account
for a complete assessment of the network.
Figure 14 also reflects the impact of the $FFCO_2$ emissions used to simulate the $CO_2$ fields.
In the 1.3-km WRF-Hestia run, there are no areas covered by more than six sites, while
the 1.3-km WRF-Vulcan run shows 39.8 % of $FFCO_2$ emissions over the LA megacity to
be covered by more than six sites. Additionally, the distribution appears nearly normal
for the 1.3-km WRF-Vulcan run. A similar discrepancy is seen between the 4-km WRF-
Hestia and WRF-Vulcan runs.  These differences between the WRF-Hestia and WRF-
Vulcan runs further highlight the importance of using the high-resolution $FFCO_2$
emissions product for the urban $CO_2$ simulation.

## 12  7    Conclusion

A set of WRF configurations varying by PBL scheme, urban surface scheme, and model
resolution has been evaluated by comparing the PBL height determined by aircraft
profiles and ceilometer, wind speed and wind direction measured by radar wind profiler,
and surface atmospheric states measured by NWS stations. The results suggest that, there
is no remarkable difference between the 4-km and 1.3-km resolution simulations in terms
of atmospheric model performances in horizontal, but the 1.3-km model runs resolve the
vertical gradients of wind fields and PBL height somewhat better as demonstrated. The
model inter-comparisons show the model using MYNN_UCM has overall better
performance than others. Coupled to $FFCO_2$ emissions products (Hestia-LA and Vulcan
2.2), a land-atmosphere modelling system was built with MYNN_UCM for studying the
heterogeneity of urban $CO_2$ emissions over the LA megacity.
The Vulcan and Hestia-LA $FFCO_2$ emission products were used to investigate the impact
of the model representation errors and emission aggregation errors on the modelled $CO_2$
concentration. Compared to the in-situ measurements during CalNex-LA, the 1.3-km
modelled $CO_2$ concentrations clearly outperform the results at 4-km resolution for
capturing both the spatial distribution and the temporal variability of the urban $CO_2$
signals due to strong $FFCO_2$ emission gradients across the LA megacity, even though no
clear improvement in the meteorological evaluation was observed across the basin. The



inter-comparison of the WRF-Hestia and WRF-Vulcan runs reinforces the importance of
using high-resolution emission products to represent correct, large spatial gradients in
atmospheric $CO_2$ concentrations for urban environments.
Based on the 1.3-km WRF-Hestia run, the coverage of the current GHG measurement
site over the LA megacity was evaluated using the modelled spatial correlations. Kort et
al. (2013) concluded a network of eight surface observation sites provided the minimum
sampling required for accurate monitoring of $FFCO_2$ emissions in LA using Vulcan at 4-
km model resolution. In this study, however, using Vulcan $FFCO_2$ emissions tend to
overestimate the observational constraint spatially, suggesting that the information lies in
multiple fine-scale plumes rather than a single urban dome over the Los Angeles basin.
Thanks to the much finer-resolution model and $FFCO_2$ emission product Hestia-LA, the
coverage of each observation site seems constrained to a more limited area. Using a high-
resolution emission data product and a high-resolution model configuration is necessary
for accurately assessing the urban measurement network.

## 8   Author contributions

S. Feng and T. Lauvaux designed the model experiments, evaluated the model
performance, and developed the assessment of the measuring network; S. Newman
provided the calibrated $CO_2$ measurements and the support for the model evaluations. P.
Rao, R. Patarasuk, D. O'Keeffe, J. Huang, Y. Song, K.R. Gurney developed and prepared
the Vulcan and Hestia emission products; R. Ahmadov contributed to the developments
of the WRF-VPRM model and relevant guideline; A. Deng provided quality control to
the observations from the National Weather Stations; L.I. Díaz-Isaac tested PBL
algorithms; S. Jeong and M.L. Fischer provided the background $CO_2$ concentration for
the LA megacity (region); R.M. Duren, C. Gerbig, Z. Li, C. E. Miller, S. Sander, K.W.
Wong, and Y. Yung provided comments and discussed the results of the study.

## Acknowledgements



A portion of this work was performed at the Jet Propulsion Laboratory, California
Institute of Technology, under contract with NASA. The Megacities Carbon Project is
sponsored in part by the National Institute of Standards and Technology (NIST). S.
Newman acknowledges funding from the Caltech/JPL President & Director's Research
and Development Fund.    K. R. Gurney thanks NIST grant 70NANB14H321.R.
Ahmadov was supported by the US Weather Research Program within the NOAA/OAR
Office of Weather and Air Quality. S. Jeong and M.L. Fischer acknowledge the support
by the Laboratory Directed Research and Development Program, Office of Science, of
the US Department of Energy under Contract No. DE-AC02-05CH11231.  Thanks to W.
Angevine at NOAA for radar wind profiler data, K. Aikin at NOAA for Aircraft WP-3D
data,    and    B.    Lefer    at    University    of    Houston    for    ceilometer    data.





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



Table 1. Common elements of the WRF-Chem configuration used in all runs.

| Option | Description |
| --- | --- |
| Microphysics | WSM5 (Hong et al., 2004) |
| Longwave radiation | RRTMG (Iacono et al., 2008) |
| Shortwave radiation | RRTMG (Iacono et al., 2008) |
| Land surface | Noah land surface model (Chen and Dudhia, 2001) |
| Cumulus scheme | Grell-3 (Grell and Dévényi, 2002) applied to 12-km domain (d01) only |



Table 2. WRF configurations used for the sensitivity runs.

| Configuration | PBL scheme | Urban surface scheme | Grid spacing (km) |
|---|---|---|---|
| BouLac_BEP_d02 | BouLac | BEP | 4 |
| BouLac_BEP_d03 | BouLac | BEP | 1.3 |
| BouLac_UCM_d02 | BouLac | UCM | 4 |
| BouLac_UCM_d03 | BouLac | UCM | 1.3 |
| MYJ_d02 | MYJ | None | 4 |
| MYN_d03 | MYJ | None | 1.3 |
| MYJ_UCM_d02 | MYJ | UCM | 4 |
| MYJ_UCM_d03 | MYJ | UCM | 1.3 |
| MYNN_d02 | MYNN | None | 4 |
| MYNN_d03 | MYNN | None | 1.3 |
| MYNN_UCM_d02 | MYNN | UCM | 4 |
| MYNN_UCM_d03 | MYNN | UCM | 1.3 |



Table 3. Locations of the 2015-era GHG measurement sites in the model domain

| Code* | Name | Type | Lat. (° N) | Lon. (° E) |
|---|---|---|---|---|
| GH | Granada Hills | Tower | 34.28 | -118.47 |
| Pasadena | Pasadena | Building top | 34.14 | -118.13 |
| MWO | Mt. Wilson | Mountain top | 34.22 | -118.06 |
| USC | University of South California | Building top | 34.02 | -118.29 |
| Compton | Compton | Tower | 33.87 | -118.28 |
| CSUF | California State University, Fullerton | Building top | 33.88 | -117.88 |
| Ontario | Ontario | Tower | 34.06 | -117.58 |
| SB | San Bernardino | Tower | 34.09 | -118.35 |
| Dryden✦ | Dryden | TCCON | 34.95 | -117.89 |
| VV | Victorville | Tower | 34.61 | -117.29 |
| UCI | University of California, Irvine | Building top | 33.64 | -117.84 |
| SCI | San Clemente Island | Tower | 32.92 | -118.49 |
| PV | Palos Verdes | In-situ non-standard | 33.74 | -118.35 |

✣La Jolla site is operating but not included in this paper

*Codes used in this paper

✦ In the analysis, we assume Dryden site is a near-surface point measurement like other sites rather than a column observation for simplicity.



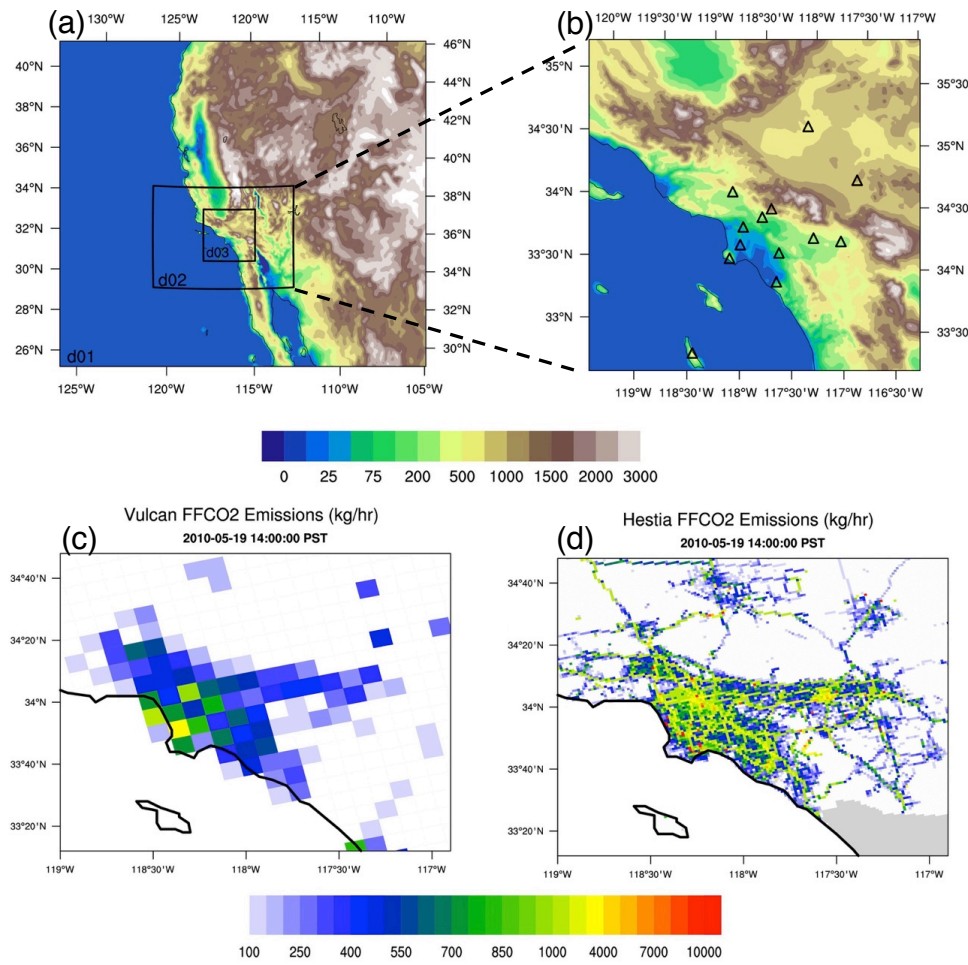

Figure 1. (a) Model domains. Contours are terrain height (unit: m).  (b) The 1.3-km
model domain (d03) and terrain height (unit: m).  Triangles represent the locations of the
GHG measurement sites. (c and d) Snapshots of the Vulcan and Hestia $FFCO_2$ emissions
(unit: kg/hr) over the LA megacity at 14:00 PST on 15 May 2010.



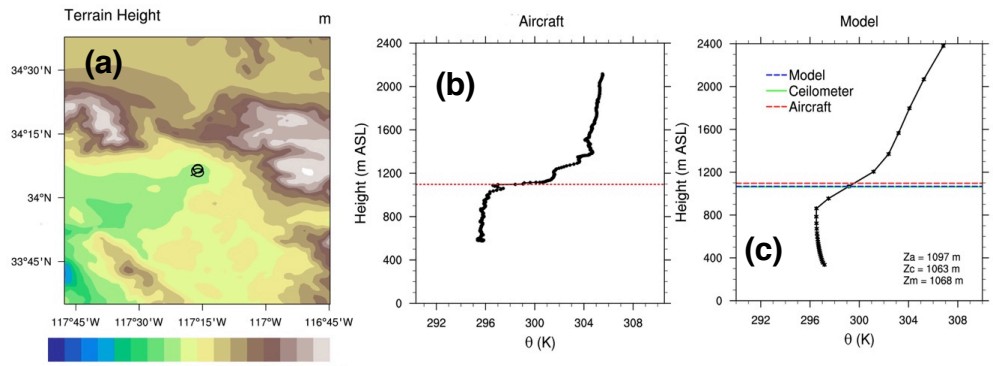

Figure 2. A case selected on 19 May 2010 at 12:25 (PST) (a) Location of the vertical
profile flown by the CalNex aircraft and the neighbouring terrain heights (units: m). (b)
In-situ potential temperature profile measured by the aircraft . The red dashed line at
~1100 m is the PBL height calculated based on the vertical gradient of potential
temperature $\Theta$(K). (c) Modelled potential temperature profile from the
MYNN_UCM_d02 configuration. The red dashed line is the aircraft-determined PBL
height ($Z_a$ in masl). The solid green line is the PBL height measured by the Caltech
ceilometer ($Z_c$ in masl). The blue line is the modelled PBL height ($Z_m$ in m).



Figure 3. Absolute difference between the aircraft-determined and modelled PBL height
for each profile: P01, P02, …, and P07 (blue bars). The pink bars in the last column
represent the averaged bias over all of the profiles for each configuration. Note that the
shorter the bar is, the better agreement the model has with the observations.



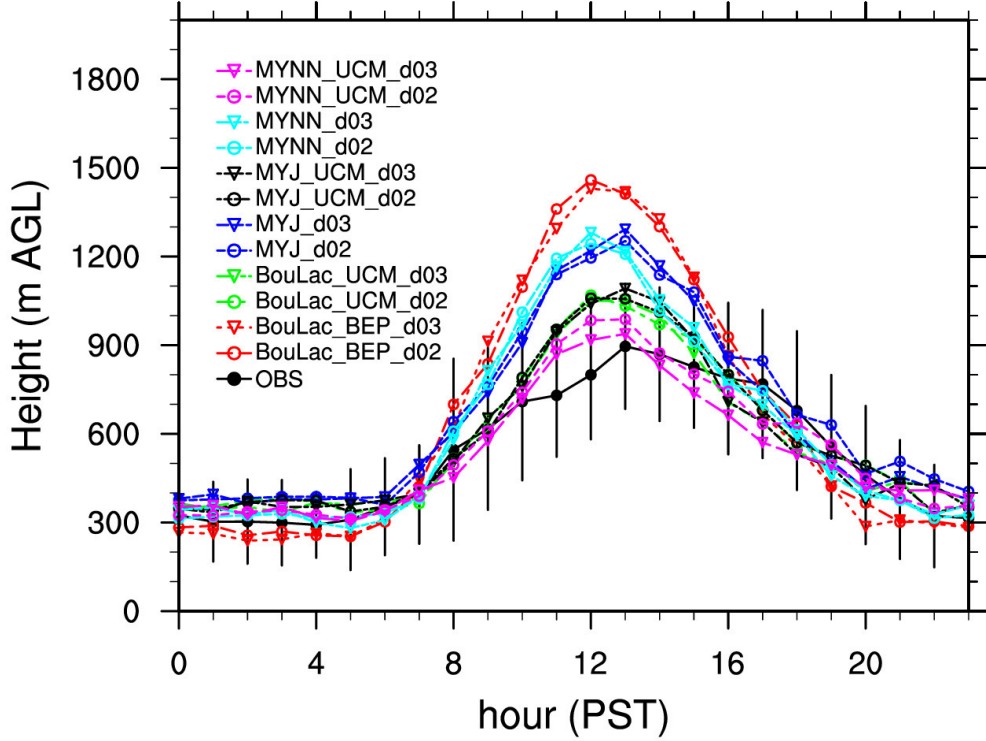

Figure 4. Average diurnal variation of the ceilometer-measured and modelled PBL
heights at California Institute of Technology (Caltech) in Pasadena, CA during 15 May
through 15 June 2010. Error bars indicate standard deviations of the means of the
ceilometer measurement.





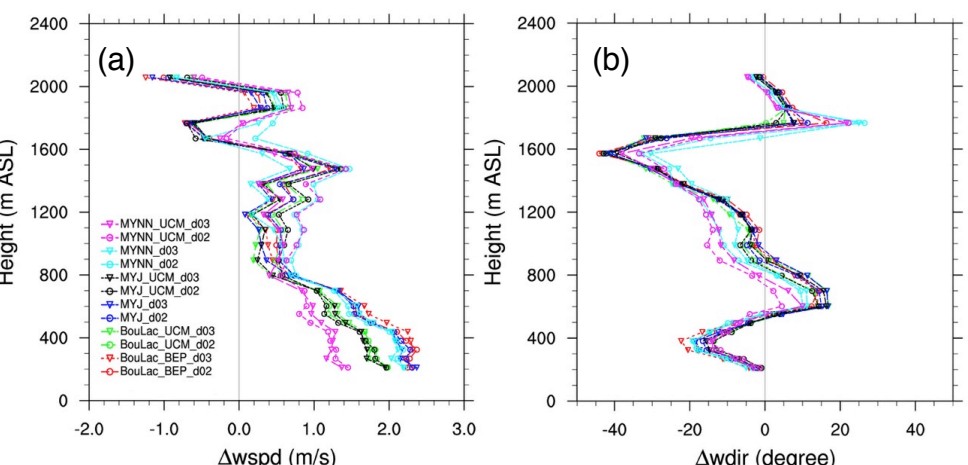

3   Figure 5. Average differences of wind profiles between the simulations and observations

4   (model – wind radar profiler) at the Los Angeles International Airport (LAX). (a) The

5   difference for wind speed (unit: m/s); (b) for wind direction (unit: degree).




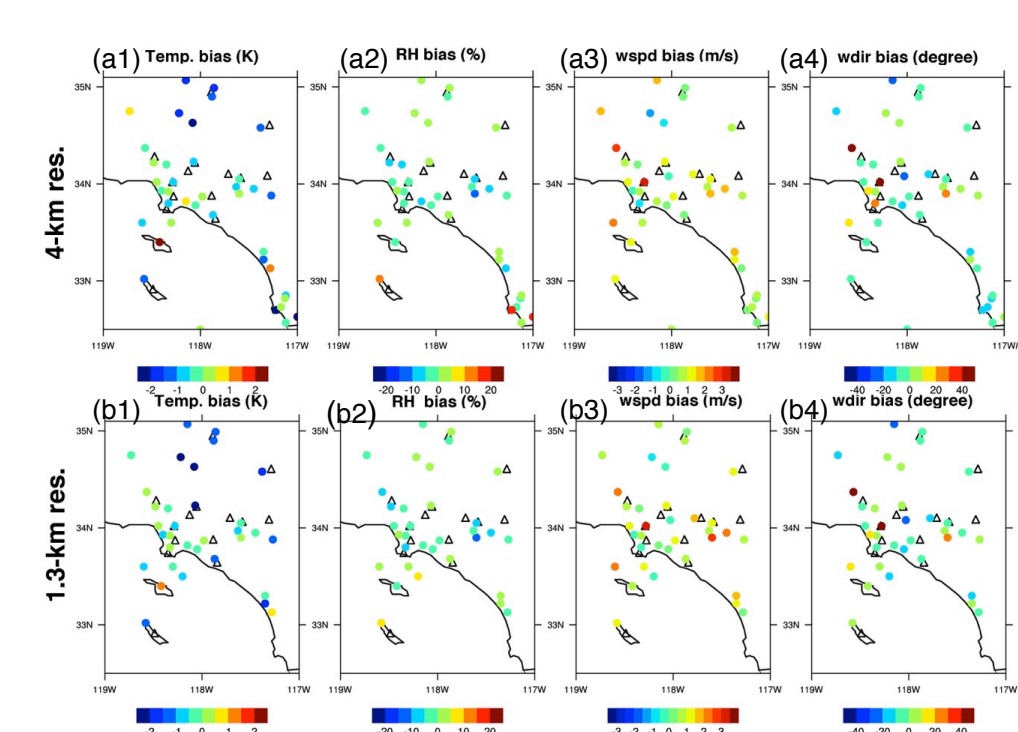

Figure 6. Bias maps of the MYNN_UCM runs versus National Weather Stations (NWS)
over the LA megacity (Model – NWS): (a1-a4) 4-km run; (b1 – b4) 1.3-km run.  Black
triangles indicate the locations of the GHG measurement sites.



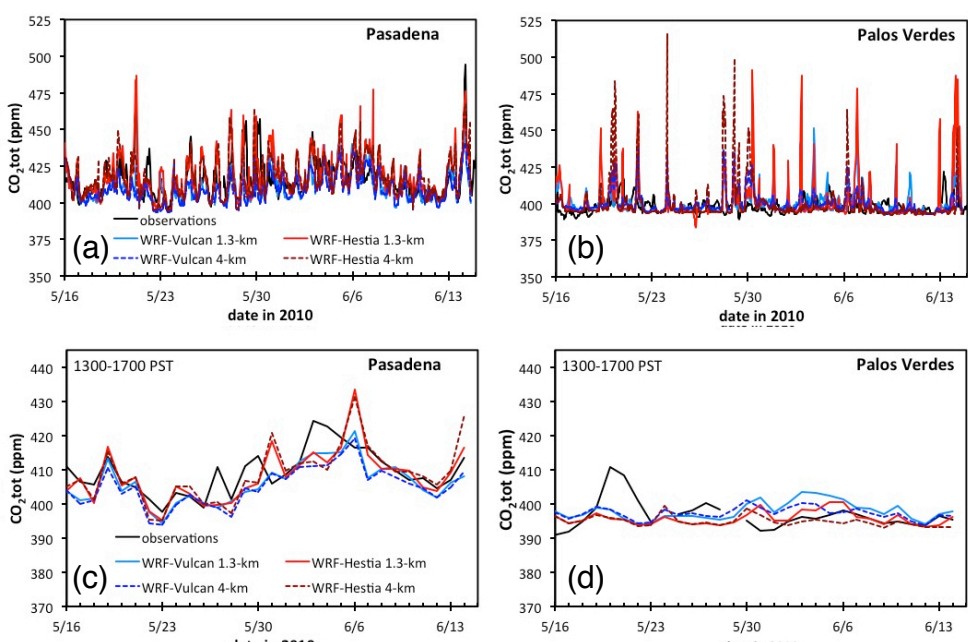

3    Figure 7. Comparison of the observed and modelled $CO_2$ concentrations at the (a and c)

4    Pasadena and (b and d) Palos Verdes sites: (a and b) is hourly time series, (c and d) is

5    daily afternoon average over 1300 – 1700 PST.



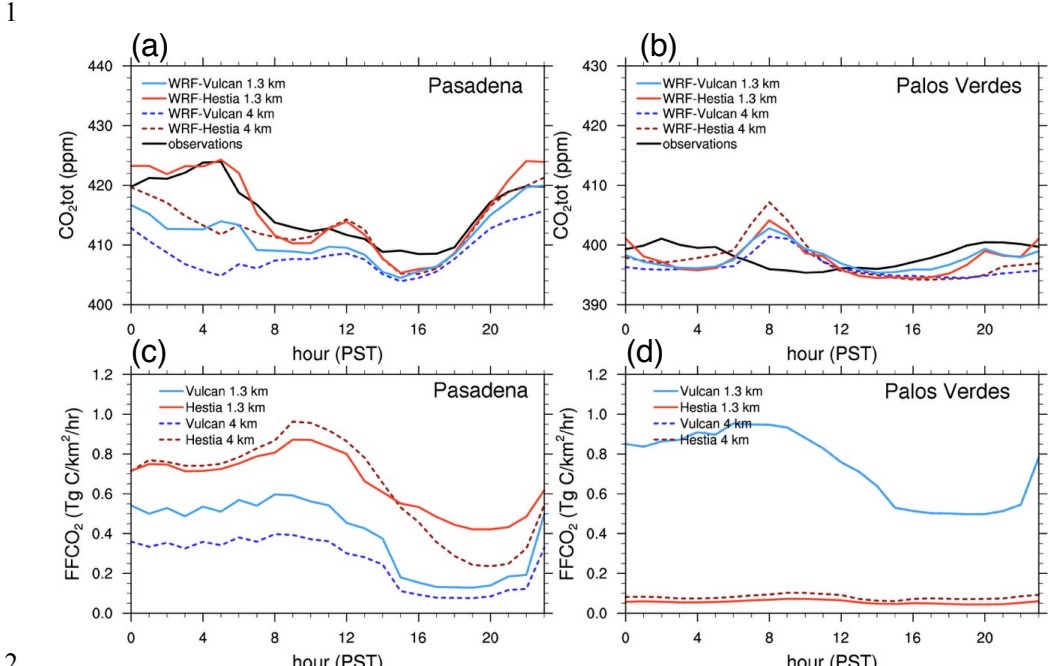

Figure 8. Averaged diurnal variation of observed and modelled $CO_2$ concentration and

FFCO$_2$ emissions for the (a and c) Pasadena and (b and d) Palos Verdes sites during

CalNex-LA.



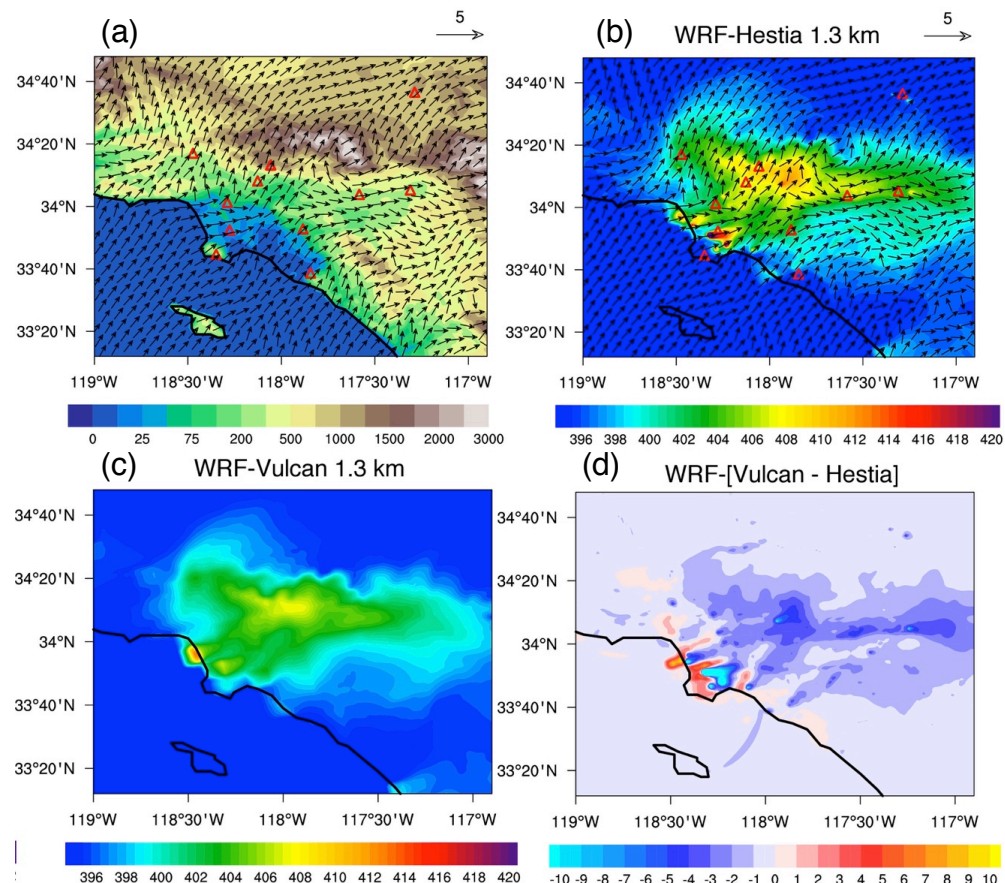

Figure 9. (a and b) The first empirical orthogonal function (EOF 1) for the surface wind pattern simulated by MYNN_UCM_d03 at 1400 PST during CalNex-LA. EOF 1 accounts for 48.1 % of the variance in the average winds. Contours: (a) terrain height (unit: m); (b) the modelled surface $CO_2$ concentration (unit: ppm) from the 1.3-km WRF-Hestia run. The red triangles indicate the locations of the GHG measurement sites. (c) The modelled $CO_2$ concentration from the 1.3-km WRF-Vulcan run (unit: ppm). (d) The difference of the modelled $CO_2$ concentration between the 1.3-km WRF-Hestia and WRF-Vulcan runs (unit: ppm).









Figure 10. The spatial correlation map (R) of the 1.3-km WRF-Hestia simulated $CO_2$
concentration between each site and the remainder of the domain at 1400 PST during the
CalNex-LA campaign. The correlation map was constructed by calculating the
simultaneous correlation of the site $CO_2$ to the $CO_2$ over rest of the LA megacity. Note
that only those pixels that pass the *t*-test at the significance level of 0.01 ($|R| \geq 0.46$) are
coloured.



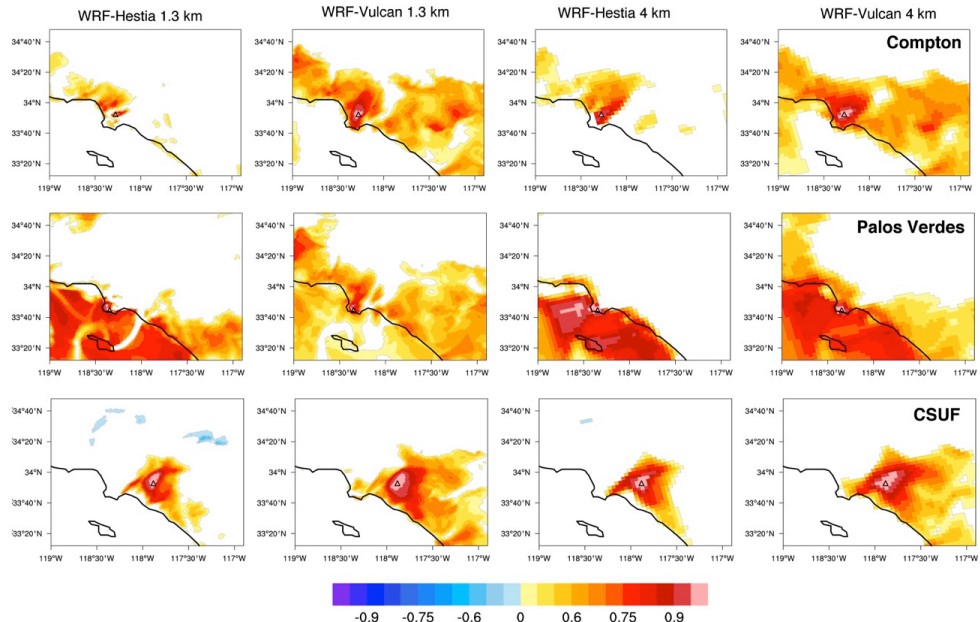

Figure 11. Same as Figure 10 but for the Compton (upper row), Palos Verdes (middle
row), and CSUF (lower row) sites only. Shown are the correlation maps of these three
measurement sites for the 1.3-km WRF-Hestia (first column), 1.3-km WRF-Vulcan
(second column), 4-km WRF-Hestia (third column), and 4-km WRF-Vulcan runs. Note
that only those pixels that pass the *t*-test at the significance level of 0.01  ($|R| \geq 0.46$) are
coloured.



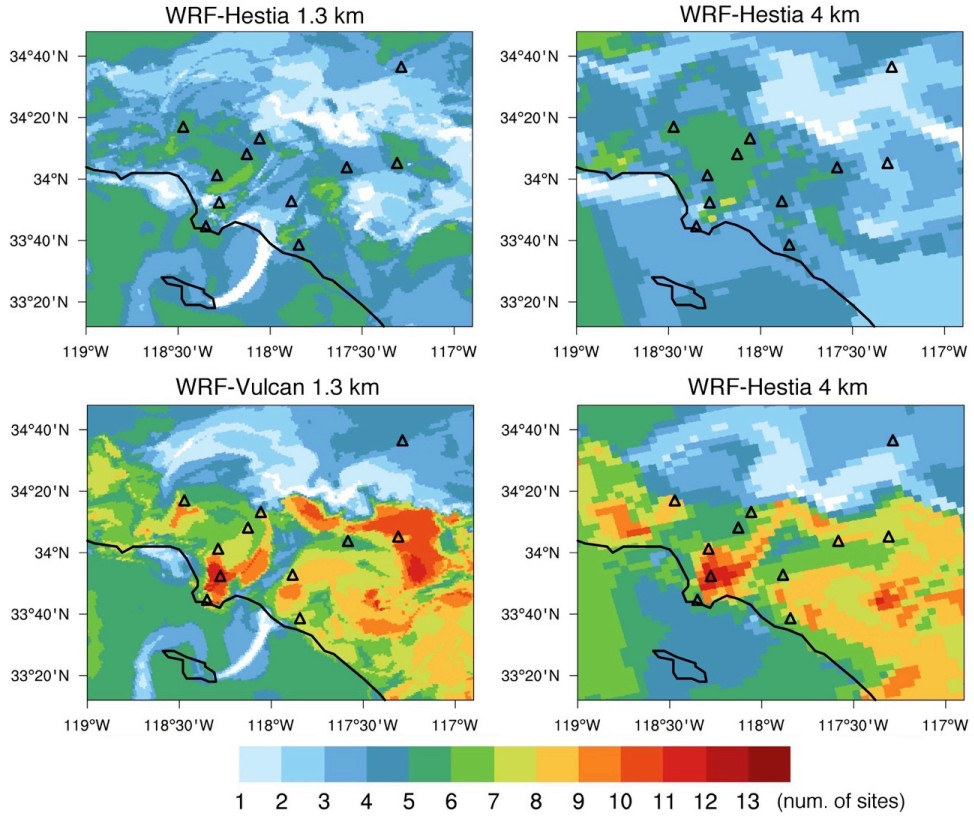

Figure 12. The composite maps of spatial correlation (R in Figure 10 and 11) for the 1.3-
km WRF-Hestia, 1.3-km WRF-Vulcan, 4-km WRF-Hestia, and 4-km WRF-Vulcan runs.
The composite map was constructed by determining the number of the observation sites
for which |R| is greater than 0.46 at each grid cell. |R| = 0.46 is the critical value at the
significance level of 0.01 of $t$-test. Specifically, white cells indicate that no sites are
correlated well at the location; dark red cells indicate that over 13 sites have good
correlation at the location. The SCI and Dryden sites are not shown on these maps.



Figure 13. Comparisons of flask-sampled and modelled (a-d) anthropogenic fossil fuel and (e-h) biogenic $CO_2$ concentration. Left column: hourly time series. The horizontal error bars on the flask-sampled data points indicate the range of dates combined in each sample. Note that much of the time periods for the $\Delta^{14}C$ samples at the Palos Verdes site



1    are before or after the modelling period. Right column: Averages at 1400 PST during

2    CalNex-LA. See Newman et al. (2015) for details about the sites and sampling

3    information.





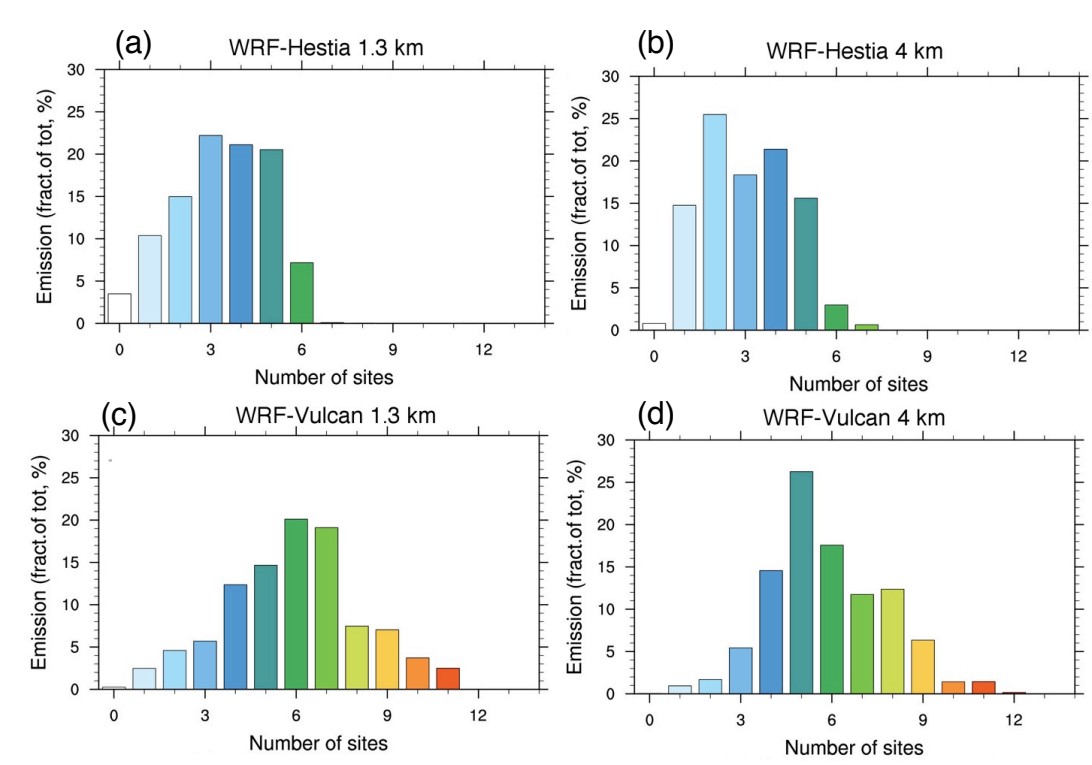

Figure 14. The fraction of the FFCO$_2$ emission over the LA megacity as function of the number of the GHG measurement sites that covers the area for (a) 1.3-km WRF-Hestia, (b) 4-km WRF-Hestia, (c) 1.3-km WRF-Vulcan, and (d) 4-km WRF-Vulcan runs during CalNex-LA.