# Peer review of "LA Megacity: a High-Resolution Land-Atmosphere"

_Atmospheric Chemistry and Physics, 2016_

## Short Comment (SC1) · 6 Apr 2016

A tremendous work for the CO2 simulation over LA megacity has been well reported. While reading, I am so interested in it, and I would like to add some comments for further improvements.

Line 18-23: The authors wrote that WRF-Chem was modified without detail description, and they only referred to Mahadevan et al. (2008), in which only VPRM is described. Would you please add more detail about the modification of WRF-Chem?

In addition, WRF-VPRM would be very sensitive to its VPRM parameters especially for nighttime respiration as Pillai et al. (2009) also showed. I wonder authors made

or modified new parameters for each vegetation class over the study domain. They may want to explain it. A table containing VPRM parameters could be good, and it will surely be a very useful for the future studies done by others. If the values were modified comparing with the default numbers, please explain the method to get the parameters.

Some figures must be corrected as below: Figure 8(d) has no dashed line for Vulcan 4km. Is it overlapped? Figure 12: Subtitles of each panel seems wrong. There are two "WRF-Hestia 4km"s. Figure 13(e): How come WRF-VPRM 1.3 km shows less temporal resolution? Please double check the legend and describe the reason please.

Thanks for the good work.

---

## Short Comment (SC2) · 7 Apr 2016

Very nice work and well-thought out paper overall. It would be nice if the inlet heights were included in the table of station details. I would also be interested in more discussion of the model-measurement agreement at the different stations and the overall skill of the model.

---

## Referee Comment (RC1) · Anonymous Referee #2 · 8 Apr 2016

General Comments

'The paper of Feng et al. entitled 'LA Megacity: a high-resolution land-atmosphere modelling system for urban CO2 emissions' compares different model resolutions and emission maps to identify optimal configurations for simulating CO2 fields over a megacity. Although this concept of comparing different models or model configurations is not new, urban air quality poses some additional challenges that the authors try to address in this paper. Additionally, they pay attention to monitoring requirements and their new network design methodology can certainly prove useful, also to estimate footprints. However, I believe the authors could stress more the importance and novelty of their study in the context of recent studies, as the summary of current literature lacks

an overview of knowledge gaps/remaining challenges and how their study fits into this (except for the paragraph about studies that focused on LA). Other than that I thank the authors for their very nice work.

Specific comments

Why have the authors decided to use one-way nesting? What would be the advantage compared to two-way nesting and what are the consequences?

The authors have chosen to simulate a two-month period per day, rather than doing the whole period in one simulation. This requires re-initialisation of the concentration fields for each day. How do the authors ensure conservation of mass between the simulations? Could you show that this re-initialisation has no impact on the simulated mass fractions?

Could the authors clearly specify whether the temporal variations for both emission product are equal? If not, how do they differ and what would be the consequence for the comparison of the products?

The authors state that for the MYNN_UCM configuration the PBL height is better represented for d03 than for d02 and that this is also reflected by other configurations. However, it appears from figure 4 that for some configurations d02 is actually better during the afternoon. This requires some reflection in the text.

Are the biases shown in Figure 6 for the whole period, including night time? If so, how do the authors reach the conclusion that the dryness in the model causes a lower PBL height (Figure 4) in the afternoon, while the PBL height is actually higher a bit earlier during the day? I would like to see a clear explanation for this, as generally I would think that dryness would cause a higher PBL height.

Page 15, ln. 23-24: 'However, during daytime, with well-mixed conditions, the discrepancy between the WRF-Hestia and WRF-Vulcan runs becomes smaller.'; and similarly: Page 16, ln. 15-17: 'For the same reason, we show that FFCO2 emissions do not play

a dominant role around 1400 PST unless there are strong local signals...'. This is an interesting note. Usually, well-mixed daytime concentrations are sampled for inverse modelling, as these conditions are usually better represented by models. That leads to the question how well we could estimate posterior fluxes if a 40% increase in FFCO2 emissions only leads to an increase of less than 1% in the total CO2 concentration (which is a rough estimate from your Figure 8 at 1400 PST using both 1.3 km simulations). Could the authors digress a bit on the consequences of this note for inverse modelling?

Section 5 introduces a new network design method. Although mentioned before that this would be discussed, I would like to see a few sentences discussing the need for such new method and the limitations of other methods. Currently, this is only briefly mentioned in the discussion. Could the authors also make a recommendation on which method would be most suitable for future use?

Technical corrections

In Section 3.1 the authors list five criteria for profile selection. The difference between point 4 and 5 should be made more clear.

In Section 3.4, the third paragraph, the authors mention the temperature difference between Granada Hills and downtown LA in °F. I would suggest to use Kelvin to make comparison with the other temperature results in Kelvin easier.

In Section 5, please mention clearly whether you used any data selection or that all data was included for the correlation maps.

The discussion now starts with new results based on flask samples of radiocarbon. Please move this to the results section. Also I would suggest to introduce the use of radiocarbon earlier, as this not mentioned previously in the paper.

---

## Referee Comment (RC2) · Anonymous Referee #1 · 17 Apr 2016

**Review of Feng et al. : LA megacity : a high-resolution land-atmosphere modelling system for urban CO2 emissions**

*Overview: The manuscript presents simulated carbon dioxide fields for 2 months centered over Los Angeles. The work demonstrates and tests the ability of a high-resolution meso-scale model to reproduce observed meteorological and carbon dioxide dynamics, with a focus on urban areas, LA in particular. The paper presents a valuable modelling approach in order to understand the temporal and spatial variability of weather variables and CO2 mixing ratio in urban and background sites. This work is appropriately placed in ACP, and contributes to the burgeoning area of studying carbon emissions from urban areas. I have some general and specific concerns delineated below, that need to be addressed before its publication.*

*General Comments: Overall things look quite nice and interesting, but I have a couple of reservations that require more explanation and must be addressed. There needs to be better presentation of modelled vs observed fields in terms of table of scores and 1:1 plots. As currently presented, it is difficult to assess model performance. The second point is that discussions on the physical reasons why a parametrized scheme is better, or on the performance of the modelling, are missing. The last parts that study correlations of the simulated CO2 fields with GHG measurements is interesting, and well oriented to further inverse modelling studies. I do not have specific remarks on this part.*

1) ***CO2 initial and boundary condition****. This is only briefly touched upon in section 2.1, and it is unclear. From what I understand the model is initialized and coupled with CO2 concentrations coming from observations. The simulations run for 36h. Do you use the predicted CO2 field from the end of the previous day to start the following day ? Or do you only use CO2 observations at the beginning of each run ? In the 2nd case, what is the spin-up time ? Is there a significant horizontal and vertical variability in the CO2 observations ? What impact do varying boundary condition choices make on simulations? We know that in regional studies boundary conditions play a tremendously important role (Lauvaux et al. TELLUS 2012). The authors must better described what they've done for boundary conditions, and make quantitative assessments of impacts of boundary condition choices on simulations.*

2) *As a large part of the simulated domains is on the sea, and as LA is largely influenced by maritime air masses, is it not a problem to ignore ocean fluxes ? Classically, **oceanic CO2 fluxes** are parameterised following Takahashi et al. (1997). A sensitivity test with ocean parametrized fluxes would be appreciated.*

3) *One objective of the paper is to assess the **PBL schemes**, but they are not physically described and the differences between the schemes are not presented . Therefore the conclusions are only limited to WRF technical configuration and physical aspects are not adressed. The 3 PBL schemes have to be described properly (closure, mixing lengths ...) to highligth the differences. Then strengths and weaknesses of each scheme need to be highlighted relating to their characteristics.*

4) *In the same way, 2 **urban surface schemes** are tested without having presented their physical differences. The scientific interest is therefore limited. We need to know the scientific reasons why UCM seems better.*

5) In the **comparison to aircraft PBL height**, the method to determine PBL height is based on the vertical virtual potential temperature gradient. Among the existing methods to determine this parameter (Ri number, parcel method ...), none is perfect. What is the impact of the choice of the method on the results ? For the 3 PBL schemes, biases on PBL heights are significant : errors of 160m in PBL height are not small by any measure. You can see for instance Riette and Lac (2016) for evaluation of PBL height over 1 year with an operation NWP model, with more satisfying values. Qualitative statements should be toned down. What is the error standard deviation ? Figure 3 is not appropriate as only biases are represented without standard deviation, and without length scale. How do you also explain that biases are smaller at 4km than at 1.3km, and that the results are different than the comparison to ceilometer ?

6) **Dynamics** : why do you use one-way nested domains and not 2-way ? Advection and temporal schemes should be specified in Table 1, with the time steps for the different resolutions. Page 7 line 16 : what is the height of the 1$^{st}$ level ?

7) **Comparison to radar wind profiler** : what is the period of evaluation ? Is it 2 months ? Tables of scores for wind speed and duration would be useful and easier to read than scores included in the text. Also, in Fig.5, if it is related to a 2 months period, it would be better to normalize the vertical coordinate by the PBL height.

8) **Comparison to NWS surface stations** : all the stations are not represented on Fig.S1 and the domain is not the same. As a complement to Fig.6, a table with scores for MYNN_UCM is necessary, not only with biases but also with rmse. As a complement to Fig.6, it would be useful to provide two figures with the orography and the urban fraction for 1.3km resolution, and to discuss if the scores are related to orography, urban area... At 1.3km, what is the resolution of the orography database ?

9) **Comparison to in-situ CO2** : once again, a table of scores (bias and rmse) with the 4 simulations, as a complement to Fig.7, is missing.

10) This study focuses only on **two months** of modelling and observations (May-June 2010). Conclusions thus must be quite limited, as one cannot extrapolate to generalized model performance from such a limited duration comparison, which could be particularly favourable or unfavourable. The limited duration of model/observations must be presented, and its impact on conclusions should be discussed. One element of this is discussing time/computation to simulate one-month, and whether the current model construct could be expected to run for years to compare w/ the observational record being recorded in LA & USA.

_Specific comments :_

P8 line 5 : It can be added that the coupling between mesoscale meteorological model and lagrangian particle model can be used in an operational framework to deal with accidental release (Lac et al., 2008).

Table 1 : There could be probably a mistake for shortwave radiation scheme : does RRTMG deal with SW radiation ?

*Abstract : The acronym FFCO2 is used before being presented.*

*References :*
*Riette and Lac : A New Framework to Compare Mass-Flux Schemes Within the AROME Numerical Weather Prediction Model, Boundary-Layer Meteorology, 2016, 1—29.*

*Lac, C., F. Bonnardot, O. Connan, C. Camail, D. Maro, D. Hebert, M. Rozet, and J. Pergaud, Evaluation of a mesoscale dispersion modelling tool during the CAPITOUL experiment, Meteorol. Atmos. Phys., 102, 263-287, 2008.*

---

## Author Comment (AC1) · 14 May 2016

Thanks for the encouraging comments. For the period of CalNex, only the Pasadena and Palos Verdes sites were operating. We will add the inlet heights of these two sites in the revised paper. The overall skill of the model will be also included in the revision.

---

## Author Comment (AC2) · 14 May 2016

**Response to SC1:**

A tremendous work for the CO2 simulation over LA megacity has been well reported. While reading, I am so interested in it, and I would like to add some comments for further improvements.

Line 18-23: The authors wrote that WRF-Chem was modified without detail description, and they only referred to Mahadevan et al. (2008), in which only VPRM is described. Would you please add more detail about the modification of WRF-Chem?

We did not modify WRF-Chem but used defaults for this study. To avoid confusion, we will clarify it in the revision. We will also add a table to the revised paper showing the default values of the VPRM parameters. Thanks!

In addition, WRF-VPRM would be very sensitive to its VPRM parameters especially for nighttime respiration as Pillai et al. (2009) also showed. I wonder authors made or modified new parameters for each vegetation class over the study domain. They may want to explain it. A table containing VPRM parameters could be good, and it will surely be a very useful for the future studies done by others. If the values were modified comparing with the default numbers, please explain the method to get the parameters.

We agree that VPRM is very sensitive to the parameters. Previous studies have used flux tower measurements to optimize the parameters (e.g. Hilton et al. (2013)). For CalNex, however, we had limited amount of samples to do so. Additionally, the biogenic CO2 is very small versus anthropogenic CO2 in Los Angles. Given these two reasons, the default values were used for simplicity in this paper. We appreciate your comments. We will take it into account in future work.

Some figures must be corrected as below:

Figure 8(d) has no dashed line for Vulcan 4km. Is it overlapped?

Yes, it's overlapped. The 4-km and 1.3-km Vulcan at Pasadena hold the same amount of fossil fuel emission. The original resolution of Vulcan is 10km by 10 km. Regridding the emission to 4-km or 1.3-km from one (original) grid cell won't make any difference in this case.

Figure 12: Subtitles of each panel seems wrong. There are two "WRF-Hestia 4km"s.

Thanks for capturing this error. We will correct the subtitle in the revision.

Figure 13(e): How come WRF-VPRM 1.3 km shows less temporal resolution? Please double check the legend and describe the reason please.

The legend is correct. We noticed that a large botanical garden covering 207 acres (i.e. The Huntington Library) is about 1.6 km away from the Pasadena site. For the 4-km run, this location and the Pasadena site is likely to be aggregated into one grid cell, which can explain the additional variability in the 4-km run at the Pasadena site. In contrast, these two locations can be in different grid cells in the 1.3-km run which could show different temporal evolutions. Please see Page 20 Line 24 – 30 for more details.

Thanks for the good work.

---

## Author Response (AR1)

**Response to Comments from RC1**

We appreciate the time and efforts by the editors and referees in reviewing this manuscript. We have addressed each of the concerns indicated in the review reports. Please see the one-to-one response (in blue) following the comments from the reviewer RC1. We believed that the revised version meets the journal publication requirements.

General Comments

'The paper of Feng et al. entitled 'LA Megacity: a high-resolution land-atmosphere modelling system for urban $CO_2$ emissions' compares different model resolutions and emission maps to identify optimal configurations for simulating $CO_2$ fields over a megacity. Although this concept of comparing different models or model configurations is not new, urban air quality poses some additional challenges that the authors try to address in this paper. Additionally, they pay attention to monitoring requirements and their new network design methodology can certainly prove useful, also to estimate footprints. However, I believe the authors could stress more the importance and novelty of their study in the context of recent studies, as the summary of current literature lacks an overview of knowledge gaps/remaining challenges and how their study fits into this (except for the paragraph about studies that focused on LA). Other than that I thank the authors for their very nice work.

Specific comments

Why have the authors decided to use one-way nesting? What would be the advantage compared to two-way nesting and what are the consequences?

One-way nesting allows the parent and the nest to exchange information strictly downscale. In this way, the nest solution does not feed back to the parent solution. Two-way nesting allows the information exchange bi-directionally. The nesting feedback impacts the parent domain's solution. This study evaluates the impact of the model physics and grid spacing on the model performance. In this case, one-way nesting is preferable to two-way nesting by which the 4-km model results will just be the smoothed 1.3-km model results.

The authors have chosen to simulate a two-month period per day, rather than doing the whole period in one simulation. This requires reinitialisation of the concentration fields for each day. How do the authors ensure conservation of mass between the simulations? Could you show that this reinitialisation has no impact on the simulated mass fractions?

Reinitialisation is commonly used in weather forecasts and regional modelling methods to prevent simulation drifting too much away from the truth. Running simulations for one-month long without reinitialisation is not proper. However, one should notice that the re-initialisation was only applied to modelled meteorology. The $CO_2$ fields were carried over from cycle to cycle without any re-initialization. The $CO_2$ mass therefore was conserved for the entire simulation.

Could the authors clearly specify whether the temporal variations for both emission product are equal? If not, how do they differ and what would be the consequence for the comparison of the products?

Both emission products were developed using "bottom-up" methods. Vulcan quantifies FFCO2 emissions for the entire contiguous United States (CONUS) hourly at approximately 10-km spatial resolution for the year of 2002. The temporal variations are driven by a combination of modeled activity (building energy modeling) and monitoring (power plant emissions). Hestia-LA is a fossil fuel CO2 emissions data product specific in space and time to individual buildings, road segments, and point sources covering the Los Angeles megacity domain for the years of 2011 and 2012. Hestia-LA uses much of the same information for the temporal variations except for the onroad emissions, for which local traffic data is employed as opposed to regional traffic data. Given the similarities, it is unlikely that the small difference in temporal variation could account for the spatial differences, through covariation with atmospheric transport, found here.

Given the limited in-situ GHG measurements that were available for CalNex, we mainly focused on the $CO_2$ concentration spatial differences over the LA basin caused by the different emission products used. One of the main conclusions of this study is that, driven by the high-resolution emission data product, i.e. Hestia, the model can reproduce the plumes from the point sources. On the other hand, the Vulcan run shows a more smeared-out CO2 distribution over the LA basin (Figure 9b vs. Figure 9c).

The authors state that for the MYNN_UCM configuration the PBL height is better represented for d03 than for d02 and that this is also reflected by other configurations. However, it appears from figure 4 that for some configurations d02 is actually better during the afternoon. This requires some reflection in the text.

Thanks for the suggestions. The text has been modified for reflecting this concern (Page 13 Line 30).

Are the biases shown in Figure 6 for the whole period, including night time?

Figure 6 shows the statics over daytime only. The clarification has been added in the figure caption in the revised paper. Thanks for the comments.

If so, how do the authors reach the conclusion that the dryness in the model causes a lower PBL height (Figure 4) in the afternoon, while the PBL height is actually higher a bit earlier during the day? I would like to see a clear explanation for this, as generally I would think that dryness would cause a higher PBL height.

Yes, dryness usually leads to the higher PBL height. Thanks for pointing out this error. The model overall dry the LA basin but with some exceptions, such as Pasadena area where the ceilometer was deployed, where the model actually moisteners the air. The moistness is consistent with the lower PBL the model simulated in Pasadena.

Page 15, ln. 23-24: 'However, during daytime, with well-mixed conditions, the discrepancy between the WRF-Hestia and WRF-Vulcan runs becomes smaller.'; and similarly: Page 16, ln. 15-17: 'For the same reason, we show that $FFCO_2$ emissions do not play a dominant role around 1400 PST unless there are strong local signals...'. This is an interesting note. Usually, well-mixed daytime concentrations are sampled for inverse modelling, as these conditions are usually better represented by models. That leads to the question how well we could estimate posterior fluxes if a 40% increase in $FFCO_2$ emissions only leads to an increase of less than 1% in the total $CO_2$ concentration (which is a rough estimate from your Figure 8 at 1400 PST using both 1.3 km simulations). Could the authors digress a bit on the consequences of this note for inverse modelling?

True. Well-mixed daytime concentrations are sampled for inverse modelling, as these conditions are usually better represented by models. However, it should be borne in mind that removing the upwind background value is required in atmospheric inversion (Lauvaux et al., 2012); only $\Delta CO_2$ is used in atmospheric inversion, not total $CO_2$ concentration ($CO_2$tot). How to derive $\Delta CO_2$, or, say, determine the background $CO_2$ ($CO_2$bkg), from the interested location remains challenges (e.g., Turnbull et al., 2015; Schuh et al., 2010). One of the common ways is subtracting the upwind $CO_2$ from the downwind location. Figure 8 shows the diurnal variation of $CO_2$tot. Roughly, if we consider $CO_2$ concentration at the PV site as $CO_2$bkg (396 ppm for 1.3-km WRF-Hestia and 397 pm for 1.3-km WRF-Vulcan), with 408 ppm and 405 ppm of $CO_2$tot at Pasadena, $\Delta CO_2$ for Pasadena is 12 ppm and 8 ppm for 1.3-km WRF-Hestia and 1.3-km WRF-Vulcan, respectively. In this case, the increase of $FFCO_2$ (mixing ratio) for 1.3-km WRF-Hestia vs. 1. .3-km WRF-Vulcan is about 50%, which is close to your estimation.

Section 5 introduces a new network design method. Although mentioned before that this would be discussed, I would like to see a few sentences discussing the need for such new method and the limitations of other methods. Currently, this is only briefly mentioned in the discussion. Could the authors also make a recommendation on which method would be most suitable for future use?

Thank you for your suggestions. We have added more sentences discussing the need and limitation of the correlation method in section 6. See Page 25 Line 8-18.

The new method assesses the correlation of "observed $CO_2$" with the neighbouring $CO_2$ concentration based on the forward model simulation. First of all, this method is computationally economical relative to the footprint method. Secondly, the method doesn't require adjoint models, which can avoid the complexity. Most importantly, it brings extreme flexibility without complexity for various platforms (i.e., in-situ, satellite, etc.) and especially outpaces the analysis for the dense sampling techniques, such as remote sensing dataset. Applying the footprint methods to satellite data at the regional scale modelling is extremely computationally time-consuming and complex.

However, as mentioned in the text, both transport and emissions play a role in the correlation method. The footprint method, in contrast, indicates the influence of the atmospheric transport to the location of the observation only. Hence, the correlation method is subject to overestimation of the influence area versus the footprint method, due to the complicated nature of the atmospheric integrator.

Technical corrections

In Section 3.1 the authors list five criteria for profile selection. The difference between point 4 and 5 should be made more clear.

These two criteria have been merged. See Page 12 Line 18. Thanks!

In Section 3.4, the third paragraph, the authors mention the temperature difference between Granada Hills and downtown LA in F. I would suggest to use Kelvin to make comparison with the other temperature results in Kelvin easier.

Changed. See Page 16 Line 4.

In Section 5, please mention clearly whether you used any data selection or that all data was included for the correlation maps.

There are no data used in Section 5. See Page 21 Line 21 for clarification.

The discussion now starts with new results based on flask samples of radiocarbon. Please move this to the results section. Also I would suggest to introduce the use of radiocarbon earlier, as this not mentioned previously in the paper.

The comparison with the flask samples and the introduction of radiocarbon have been moved to section 3.6 following the comparison to in-situ measured total $CO_2$. Thanks!

Reference:

Lauvaux, T., Schuh, A. E., Uliasz, M., Richardson, S., Miles, N., Andrews, A. E., Sweeney, C., Diaz, L. I., Martins, D., Shepson, P. B., and Davis, K. J.: Constraining the $CO_2$ budget of the corn belt: exploring uncertainties from the assumptions in a mesoscale inverse system, Atmos. Chem. Phys., 12, 337-354, 10.5194/acp-12-337-2012, 2012.

Schuh, A. E., Denning, A. S., Corbin, K. D., Baker, I. T., Uliasz, M., Parazoo, N., Andrews, A. E., and Worthy, D. E. J.: A regional high-resolution carbon flux inversion of North America for 2004, Biogeosciences, 7, 1625-1644, 10.5194/bg-7-1625-2010, 2010.

Turnbull, J. C., Sweeney, C., Karion, A., Newberger, T., Lehman, S. J., Tans, P. P., Davis, K. J., Lauvaux, T., Miles, N. L., Richardson, S. J., Cambaliza, M. O., Shepson, P. B., Gurney, K., Patarasuk, R., and Razlivanov, I.: Toward quantification and source sector identification of fossil fuel $CO_2$ emissions from an urban area: Results from the INFLUX experiment, Journal of Geophysical Research: Atmospheres, 120, 2014JD022555, 10.1002/2014JD022555, 2015.

**Response to Comments from RC2**

We appreciate the time and efforts by the editors and referees in reviewing this manuscript. We have addressed each of the concerns indicated in the review reports. Please see the one-to-one response (in blue) following the comments from the reviewer RC2. We believe that the revised version meets the journal publication requirements.

Overview: The manuscript presents simulated carbon dioxide fields for 2 months centered over Los Angeles. The work demonstrates and tests the ability of a high-resolution meso-scale model to reproduce observed meteorological and carbon dioxide dynamics, with a focus on urban areas, LA in particular. The paper presents a valuable modelling approach in order to understand the temporal and spatial variability of weather variables and $CO_2$ mixing ratio in urban and background sites. This work is appropriately placed in ACP, and contributes to the burgeoning area of studying carbon emissions from urban areas. I have some general and specific concerns delineated below, that need to be addressed before its publication.

General Comments: Overall things look quite nice and interesting, but I have a couple of reservations that require more explanation and must be addressed. There needs to be better presentation of modelled vs observed fields in terms of table of scores and 1:1 plots. As currently presented, it is difficult to assess model performance. The second point is that discussions on the physical reasons why a parametrized scheme is better, or on the performance of the modelling, are missing. The last parts that study correlations of the simulated $CO_2$ fields with GHG measurements is interesting, and well oriented to further inverse modelling studies. I do not have specific remarks on this part.

1) **$CO_2$ initial and boundary condition**. This is only briefly touched upon in section 2.1, and it is unclear. From what I understand the model is initialized and coupled with $CO_2$ concentrations coming from observations. The simulations run for 36h. Do you use the predicted $CO_2$ field from the end of the previous day to start the following day ? Or do you only use $CO_2$ observations at the beginning of each run ? In the 2nd case, what is the spin-up time ? Is there a significant horizontal and vertical variability in the $CO_2$ observations ? What impact do varying boundary condition choices make on simulations? We know that in regional studies boundary conditions play a tremendously important role (Lauvaux et al. TELLUS 2012). The authors must better described what they've done for boundary conditions, and make quantitative assessments of impacts of boundary condition choices on simulations.

We initialized $CO_2$ fields from the NOAA curtain dataset at the beginning of the first cycle. The simulation runs for 36 hour for each cycle with 12-hour setback for spin-up. For each cycle, only the meteorology is re-initialized; $CO_2$ fields are carried over from the last cycle. For instance, the first simulation cycle is 00 UTC 15 May to 12 UTC 16 May 2010, and the second cycle is 00 UTC 16 May to 12 UTC 17 May 2010. The initial conditions for 00 UTC 15 May include NARR, NCEP SST and NOAA curtain ($CO_2$). The initial conditions for 00 UTC 16 May include NARR, NCEP SST and WRF-modelled $CO_2$ on 00 UTC 16 May from the previous cycle. Briefly, we did not re-initialize $CO_2$ for each cycle to assure mass conservation over the model domain. The clarification for $CO_2$ IC and BC has been added in the revised paper (see Page 11 Line 27-29).

We agree that the boundary conditions (BCs) are critical for the $CO_2$ simulations. In this study, we found there is no significant horizontal and vertical variability in the NOAA curtain dataset; semi-constant BC was used. We have also applied $CO_2$ modelled by GEOS-Chem BC for our region of interest, which introduced ~+10 ppm model-data mismatch in the WRF model results. This is similar to the findings by (Lauvaux et al., 2012), who found the model-data mismatch was more than 20 ppm in summer over the corn belt area. It also reflects the challenges in determining $CO_2$ background values for regional scale simulations. We therefore end up with using semi-constant values ("NOAA Curtain") as the model BC in the paper. The NOAA Curtain dataset mainly represents oceanic clean air. In May – June, west to southwest clean marine flow prevails over the Los Angeles Megacity. Using a semi-constant dataset is fairly close to the reality, introducing lower errors to the regional, modelled $CO_2$ relative to global models, such as GEOS-Chem. However, during October to March, Santa Ana wind events occur frequently, during which easterly to north-easterly winds predominate over the LA basin, and the oceanic air is polluted. In this case, using constant values is no longer feasible.

2) As a large part of the simulated domains is on the sea, and as LA is largely influenced by maritime air masses, is it not a problem to ignore ocean fluxes ? Classically, **oceanic $CO_2$ fluxes** are parameterised following Takahashi et al. (1997). A sensitivity test with ocean parametrized fluxes would be appreciated.

The LA megacity is one of the top three fossil fuel emitters in the U.S. Roughly estimated from Hestia at the Pasadena site, the order of fossil fuel emission is about 10-20 umol/m$^2$/s. The typical oceanic $CO_2$ flux -0.15 umol/m$^2$/s (Torres et al., 2011), 0.2 umol/m$^2$/s (Mu et al., 2014), represents only 1-2% of FFCO$_2$ fluxes and even less compared to CO$_2$tot. Because of that, we have ignored the oceanic $CO_2$ signal for simplicity in this study. Yet we do agree that a sensitivity test with oceanic flux would be interesting and should be included in future work. This explanation has been added to the revised paper. See Page 11 Line 18-23.

3) One objective of the paper is to assess the **PBL schemes**, but they are not physically described and the differences between the schemes are not presented . Therefore the conclusions are only limited to WRF technical configuration and physical aspects are not addressed. The 3 PBL schemes have to be described properly (closure, mixing lengths ...) to highlight the differences. Then strengths and weaknesses of each scheme need to be highlighted relating to their characteristics.

In this study, we have selected three most commonly used TKE-driven PBL schemes for comparison, including MYJ, MYNN, and BouLac. MYJ (Janjić, 1994) determines the PBL from the TKE where the PBL top is defined as the height where the TKE profile decreases to the threshold of 0.2 m$^2$s$^{-2}$. MYNN2 (Nakanishi and Niino, 2006) is tuned to a database of large eddy simulations (LES) in order to overcome the typical biases associated with other MY-type schemes, such as insufficient growth of convective boundary layer and under-estimated TKE. Additionally, MYNN also considers sub-grid TKE terms, and it determines the PBL top as the height at which the TKE falls below $1.0 \times 10^{-6}$ m$^2$ s$^{-2}$. BouLac (Bougeault and Lacarrere, 1989) has an option designed for use with BEP multi-layer and UCM.  It determines PBL top at which TKE reaches 0.005 m$^2$ s$^{-2}$. They all are 1.5 order local closure schemes that only consider immediately adjacent vertical levels in the model, which may not fully account for deeper vertical mixing associated with larger eddies and associated countergradient flux correction terms and, thus, tends to prevent the PBL from mixing as deeply to produce cooler and moister conditions. On the contrary, the non-local closure schemes considering a deeper layer account for countergradient fluxes and, thus, generally represent deep PBL circulation better than local schemes. The PBL schemes were reviewed by Cohen et al. (2015).

The main reason that we focus on the TKE-driven PBL schemes only is that the explicitly estimated turbulence fluxes can be used to drive Lagrangian particle dispersion models to computer influence footprints for subsequent atmospheric inversions. Through the model evaluation, we aimed to determine an optimal model configuration for modelling urban $CO_2$ over the LA megacity, and eventually to use the same system for synthesis analysis in future. In this study, we concluded that MYNN in combination with UCM is optimal for the LA modelling framework, which is consistent with the findings of Coniglio et al. (2013) who showed MYNN supports deep convection springtime.

The strengths and weaknesses of each scheme with their characteristics have been added to the revised paper. See Page 9 Line 3-15. Thanks!

4) In the same way, 2 **urban surface schemes** are tested without having presented their physical differences. The scientific interest is therefore limited. We need to know the scientific reasons why UCM seems better.

UCM is a single-layer urban canopy model, representing urban geometry and 3-D urban surfaces such as walls, roofs and roads. Furthermore, the sensible heat fluxes from the surface are calculated with Monin-Obukhov similarity theory and Jurges formula. The important factor of anthropogenic heat (AH) and its diurnal profiles are included and added to the sensible heat flux from the street canyon (Chen et al., 2011). BEP allows a direct interaction between the buildings and the PBL. BEP considers the 3-D urban surface and the vertical distribute source of buildings and momentum sinks throughout the whole canopy layer. The effects of vertical and horizontal surfaces on momentum, TKE and potential temperature are included. However, BEP requires very high vertical resolution within the PBL and is only compatible with MYJ and BouLac PBL schemes. Given that BEP is computationally expensive, we only test it with BouLac in this study.

The scientific reasons to explain the urban schemes' characteristics have been added to the revised paper. See Page 14 Line 9-12. Thanks!

5) In the **comparison to aircraft PBL height**, the method to determine PBL height is based on the vertical virtual potential temperature gradient. Among the existing methods to determine this parameter (Ri number, parcel method ...), none is perfect. What is the impact of the choice of the method on the results ?

We have used the vertical virtual potential temperature gradient and Ri number methods to determine PBL top (see Figures R1 and R2 below). Compared to the vertical virtual potential temperature gradient method, the Ri method shows larger bias in the modelled PBL top, deeper for daytime, shallower for nighttime, but the overall conclusion remains the same in terms of model inter-comparison, namely MYNN_UCM shows better agreement with ceilometer measured PBL height. We therefore show only the vertical virtual potential temperature gradient determined PBL in the text.

[Figure]

Figure R1. Absolute difference between the aircraft-determined and modelled PBL height for each profile (flt_yyyymmdd, blue bars) using virtual potential temperature gradient (top) and Richardson number (bottom). The pink bars in the last column represent the averaged bias over all of the profiles for each configuration. Note that the shorter the bar is, the better agreement the model has with the observations.

[Figure]

Figure R2. Average diurnal variation of the ceilometer-measured (obs) and modelled PBL heights at California Institute of Technology (Caltech) in Pasadena, CA during 15 May through 15 June 2010. Error bars indicate one standard deviations. Upper: the vertical virtual potential temperature ($\theta_v$) gradient determined PBL. Lower: the Ri number determined PBL. Note that the ceilometer-measured PBL top (black solid line) is the same in these two panels.

For the 3 PBL schemes, biases on PBL heights are significant : errors of 160m in PBL height are not small by any measure. You can see for instance Riette and Lac (2016) for evaluation of PBL height over 1 year with an operation NWP model, with more satisfying values. Qualitative statements should be toned down. What is the error standard deviation? Figure 3 is not appropriate as only biases are represented without standard deviation, and without length scale. How do you also explain that biases are smaller at 4km than at 1.3km, and that the results are different than the comparison to ceilometer?

Please note that Figure 3 (in the manuscript) and Figure R1 (in the response) show the absolute
difference between the observation and model for each aircraft profile we selected, so the error of
m in PBL height is the mean over seven aircraft profiles only (small sample). We did not intend
to make any specific conclusion based on seven profiles. The take-home message of Figure 3 is that
the differences between the modelled and aircraft-determined PBL height differ case by case, and
none of the model physics options is systematically better than the others. To further define the
optimal physics for the PBL height simulation, we presented the all-hours statistics with the
ceilometer data in section 3.2 and Figure 4.

Given the relative large number of the ceilometer measurements, similar model evaluation (Table R1)
to that of Riette and Lac (2016) has been done and been added to the revised paper (Table 3).
Compared to the values evaluated by Riette and Lac (2016), -9.17 m for bias and 115 m for RMSE
(PMMC09), the scores of MYNN_UCM fall in a comparable range.

Table R1. Comparison Statistics of model performance relative to the  ceilometer data over
– 1700 PST (unit: m AGL)

|  | Mean | Bias | Standard deviation | RMSE |
|---|---|---|---|---|
| OBS | 835.7 | - | 223.8 | - |
| MYNN_UCM_d03 | 828.8 | -6.9 | 82.7 | 89.7 |
| MYNN_UCM_d02 | 820.4 | -15.3 | 66.1 | 94.5 |
| MYNN_d03 | 1055.6 | 219.9 | 205.8 | 278.2 |
| MYNN_d02 | 1029.4 | 193.7 | 200.0 | 254.3 |
| MYJ_UCM_d03 | 961.4 | 125.8 | 154.9 | 168.8 |
| MYJ_UCM_d02 | 971.4 | 135.7 | 109.3 | 157.7 |
| MYJ_d03 | 1115.3 | 279.7 | 174.4 | 308.7 |
| MYJ_d02 | 1105.1 | 269.5 | 150.9 | 291.6 |
| BouLac_UCM_d03 | 936.1 | 100.5 | 147.3 | 149.9 |
| BouLac_UCM_d02 | 958.7 | 123.1 | 104.8 | 148.7 |
| BouLac_BEP_d03 | 1233.9 | 398.3 | 239.0 | 442.2 |
| BouLac_BEP_d02 | 1244.3 | 408.6 | 219.5 | 446.0 |

6) **Dynamics** : why do you use one-way nested domains and not 2-way ?

One-way nesting allows the parent and the nest to exchange information strictly downscale. In this way, the nest solution does not feed back to the parent solution. Two-way nesting allows the information exchange bi-directionally. The nesting feedback impacts the parent domain's solution.  This study evaluates the impact of the model physics and grid spacing on the model performance. In our experiment, one-way nesting is preferable to two-way nesting for which the 4-km model results will just be smoothed 1.3-km model results.

Advection and temporal schemes should be specified in Table 1, with the time steps for the different resolutions. Page 7 line 16 : what is the height of the 1st level ?

5th and 3rd order differencing for horizontal and vertical advection respectively are used. 3rd order Runge-Kutta is used for time integration with 45, 24, and 5 s for outermost, middle, innermost domains, respectively. These specifications have been added to Table 1 in the revised paper. The first level of the model setup is about 8 m above ground level (see Page 8 Line 21).

7) **Comparison to radar wind profiler** : what is the period of evaluation ? Is it 2 months ? Tables of scores for wind speed and duration would be useful and easier to read than scores included in the text.

The evaluation was done over daytime for the entire one-month simulation. Our intent in this paper is to present the model errors varying with height. For this pupose a figure is preferable.

Also, in Fig.5, if it is related to a 2 months period, it would be better to normalize the vertical coordinate by the PBL height.

 We appreciate your suggestion and will take it into account in future work.

8) **Comparison to NWS surface stations** : all the stations are not represented on Fig.S1 and the domain is not the same. As a complement to Fig.6, a table with scores for MYNN_UCM is necessary, not only with biases but also with rmse. As a complement to Fig.6, it would be useful to provide two figures with the orography and the urban fraction for 1.3km resolution, and to discuss if the scores are related to orography, urban area... At 1.3km, what is the resolution of the orography database ?

Figure S1 is a map showing the location of all of the GHG measurement over the LA basin, which matches the triangles in Figure 1b, 6, 9a, 9b, and 12. Figure 1b shows the orography for the 1.3 km domain. Figure S1 shows the orography as well, although the domain is not exactly the same as other figures. Usually to the locations of NWS station relative to the location of the GHG measurements (triangles), we estimated the relevant orography. We choose to keep the figures as in the original manuscript to avoid redundancy.

We tried to explain the model bias with orography at the beginning, but could find no clear correlation. The RMSE maps below have been added to the revised paper (Figure 7).

[Figure]

Figure R3. RMSE maps of the MYNN_UCM runs versus National Weather Stations (NWS) over the LA megacity (Model – NWS): (a1-a4) 4-km run; (b1 – b4) 1.3-km run. Black triangles indicate the locations of the GHG measurement sites.

9) **Comparison to in-situ CO$_2$** : once again, a table of scores (bias and rmse) with the 4 simulations, as a complement to Fig.7, is missing.

Thanks for the suggestion. We have added the two tables (Table 4 and 5) below as complements to Figure 7 in the revised paper.

Table R2. Statistics of modelled CO$_2$ (unit: ppm) with different configurations relative to in-situ CO$_2$ between 1300 – 1700 PST

|  | Pasadena | | Palos Verdes | |
|---|---|---|---|---|
|  | bias | RMSE | bias | RMSE |
| 1.3 km WRF-Hestia | 8.91 | 18.43 | 2.57 | 17.00 |
| 4 km WRF-Hestia | 7.03 | 14.50 | 8.09 | 19.64 |
| 1.3 km WRF-Vulcan | 1.20 | 11.10 | 5.03 | 10.62 |
| 4 km WRF-Vulcan | -1.38 | 9.13 | 4.20 | 9.40 |

Table R3. Statistics of daily afternoon averaged modelled CO$_2$ (unit: ppm) with different configurations relative to in-situ CO$_2$[*]

|  | Pasadena | | Palos Verdes | |
|---|---|---|---|---|
|  | bias | RMSE | bias | RMSE |
| 1.3 km WRF-Hestia | -1.39 | 6.21 | -0.75 | 4.71 |
| 4 km WRF-Hestia | 0.58 | 4.38 | -1.77 | 4.59 |
| 1.3 km WRF-Vulcan | -3.43 | 5.51 | 1.37 | 5.21 |
| 4 km WRF-Vulcan | -4.41 | 6.12 | 0.58 | 4.38 |

*Averaged over 1300 – 1700 PST

10) This study focuses only on **two months** of modelling and observations (May-June 2010). Conclusions thus must be quite limited, as one cannot extrapolate to generalized model performance from such a limited duration comparison, which could be particularly favourable or unfavourable. The limited duration of model/observations must be presented, and its impact on conclusions should be discussed.

The Los Angeles basin is surrounded to the north and east by mountain ranges with summits of 2-3 km, with the ocean to the west and the desert to the north. From April to September, LA is in a warm, dry, and stable air mass. Alongshore steady wind flow predominates this area. In contrast, from October to March, moist onshore flows bring precipitation to LA.
Details about LA climate can be found in the study of Conil and Hall (2006).

The focus of this study is from the middle of May to the middle of June, which is representative of the dry season. We agree that the study based on a one-month simulation has its limitations. The model has to be evaluated and verified as the time period and spatial region of interest change. The limitation of this study has been added to the revised paper (see Page 26 Line 16-25).

One element of this is discussing time/computation to simulate one-month, and whether the current model construct could be expected to run for years to compare w/ the observational record being recorded in LA & USA.

This one-month high-resolution simulation with 288x288x50 grids and 5-s time steps has taken 11520 CPU hours (45 hours x 256 processors) on NAS High performance supercomputer Pleiades. See Page 26 Line 22-25. Using the same number of processors on Pleiades, a one-year simulation will take about 23 days to complete, which is still reasonable. It is, however, not practical for the large scale, i.e., the contiguous United States.

Specific comments :

P8 line 5 : It can be added that the coupling between mesoscale meteorological model and lagrangian particle model can be used in an operational framework to deal with accidental release (Lac et al., 2008).
Added. See Page 9 Line 18-20.

Table 1 : There could be probably a mistake for shortwave radiation scheme : does RRTMG deal with SW radiation ?
The RRTMG shortwave scheme has been included in version 3.1 and above.

Abstract : The acronym $FFCO_2$ is used before being presented.

Thanks for catching this. The full name has been added in the revised paper.

[revised manuscript text omitted]